# Two-layer homolog network approach for PFAS nontarget screening and retrospective data mining

Zhaoyu Jiao[1], Sachi Taniyasu [2], Nanyang Yu [1], Xuebing Wang[1], Nobuyoshi Yamashita[2] & Si Wei [1] ✉

The rapid increase of novel per- and polyfluoroalkyl substances (PFAS) raises concerns, while their identification remains challenging. Here, we develop a two-layer homolog network approach for PFAS nontarget screening using mass spectrometry. The first layer constructs networks between homologs, with evaluation showing that it filters 94% of false candidates. The second layer builds a network between classes to expedite the identification of PFAS. We detected 94 PFAS in twelve waterproof products and two related industrial sludges, including 36 novel PFAS not previously reported in any sample. A local dataset is constructed for retrospective analysis by re-analyzing our previous samples, revealing fifteen novel PFAS in samples collected in 2005. The retrieval of the public database MassIVE uncovers novel PFAS in samples from seven countries. Here, we reveal the historic and global presence of novel PFAS, providing guidance for the management and policy-making concerning persistent chemicals.

Per- and polyfluoroalkyl substances (PFAS) have been widely used in industrial applications for decades due to their exceptional properties, resulting in their ubiquity in the environment[1,2] and in humans[3,4]. The remarkable stability of the carbon-fluorine bond renders PFAS persistent, thereby inflicting continuous harm to human health and ecosystems globally[5]. Legacy PFAS, including perfluorooctane sulfonic acid (PFOS) and perfluorooctanoic acid (PFOA) have been globally banned from production and use due to their hazardous nature[6]. Numerous PFAS alternatives have been developed, consumed, and released into the environment[7,8]. More than seven million PFAS-related structures are cataloged in the PubChem database[9], rendering the assessment and regulation of PFAS a formidable challenge.

Reports on novel PFAS typically lag far behind their production and emission. For instance, 6:2 chlorine substituted perfluoroalkyl ether sulfonic acid was used for 30 years before being reported[10]. P-perfluorous nonenoxybenzenesulfonate has been manufactured since the 1980s and was reported recently[11]. These novel PFAS are alarming because they exhibit toxic effects comparable to PFOS[12–15]. However, there are still countless novel PFAS that have eluded monitoring, according to the results of extractable organofluorine[16,17]. High-resolution mass spectrometry and nontarget analysis enable the identification of novel PFAS without prior information. Thousands of novel PFAS have been identified through nontarget analysis[18].

Mass defect, Kendrick mass defect analysis, and chemical formula assignments are commonly used nontarget screening methods for PFAS[19–21]. However, these methods only consider the mass of the precursor ions, resulting in high false positive rates in complex environmental samples. The collision cross section versus m/z values[22] and m/C ratios[23] introduce additional dimensional information, which can partially reduce false positives. Diagnostic fragments, fragment differences, neutral losses, and fragment ion flagging introduce MS/MS spectral information to prioritize PFAS[24,25]. Nevertheless, these features in MS/MS spectra are summarized from known PFAS, which may limit the identification of novel PFAS with different fragment patterns. In addition, structure diagnosis remains a major challenge in nontarget analysis. The molecular network based on spectra similarity clustering may provide new insights into the analysis of PFAS. Recently, several environmental studies have used molecular networks to screen for

[1]State Key Laboratory of Pollution Control and Resource Reuse, School of the Environment, Nanjing University, Nanjing, People's Republic of China. [2]National Institute of Advanced Industrial Science and Technology (AIST), 16-1 Onogawa, Tsukuba, Ibaraki, Japan. ✉e-mail: weisi@nju.edu.cn

transformation products based on their structural similarity with parent compounds[26,27]. The approach combining homolog screening using MS1 information, and molecular networks using MS/MS information, could complement each other for the nontarget analysis of PFAS.

Besides, current research on novel PFAS focused on samples collected from specific times or locations, lacking the distribution of PFAS on large spatiotemporal scales. This limitation hinders effective assessments of the potential hazards posed by novel PFAS. However, it is a challenging task for researchers to collect samples at different temporal scales, from national to global levels. The MassIVE platform reposit millions of mass spectrometry raw files derived from a variety of samples, including air, water, soil, plants, animals, and humans, collected globally[28]. Specialized search tools like MASST and ReDU have been developed for efficient exploration of this extensive repository based on the accurate mass and MS/MS data[29,30]. Through these tools, Gentry et al. have discovered microbial bile amidates associated with inflammatory bowel disease[31]. The utilization of large-scale mass spectrometry data could provide new perspectives for retrospective analysis.

Extensive target studies on PFAS in waterproof-related products, particularly textiles, have been conducted[32,33]. Recently, several nontarget studies have also explored the presence of emerging PFAS in textile wastewater and waterproof paper products[34,35], highlighting that waterproof-related products may serve as notable sources of emerging PFAS in the environment. In this work, we conducted nontarget mass spectrometry on waterproof products and related industrial sludges to identify novel PFAS. The samples analyzed included 2 waterproof chemical samples, 4 rubber car wiper samples, 5 textile samples, 1 waterproof cloth sample, and 2 industrial sludge samples. A two-layer homolog network approach was developed for accelerating analysis and assisting identification of PFAS. The first layer internal network integrates homolog screening

and molecular networking, effectively removing 94% of false positive features. The second layer network is constructed based on the similarity of class spectra, clustering structurally similar classes together for simultaneous identification. A total of 94 PFAS were identified, including 36 PFAS that have not been previously reported. A local dataset was then constructed to investigate the historical exposure of novel PFAS. A global database retrieval was conducted and novel PFAS were found in samples from around the world. Our study suggests the potential historical and global exposure risks of previously unreported PFAS.

## Results

### Two-layer homolog network

We developed a two-layer homolog network (Fig. 1a) for the nontarget analysis of PFAS, with specific details provided in the "Methods" section. Briefly, precursor ions differing by $CF_2$ units were selected as candidates through homolog screening (Fig. 1b)[36]. After spectra filtering, the first-layer internal network was constructed for each candidate class based on GNPS spectra similarity (similarity > 0.3, Fig. 1c)[37]. The spectra of each class were then merged into a single spectrum, and the second-layer external network was constructed based on class spectra similarity (similarity > 0.2, Fig. 1d). The nodes in the external network were grouped into communities based on the Louvain community detection algorithm[38], with the resolution parameter set to 1.0.

Our internal network which integrates homolog screening and molecular network while utilizing both MS1 and MS/MS information, could reduce false positives in homolog screening. We evaluated the denoising capability of the internal network using the local spectra database and an in-house standard sample. Utilizing the local spectra database, we observed that the internal homolog network demonstrated a 94% reduction in false positive features compared to homolog screening (Supplementary Fig. 1a). The false negative rate

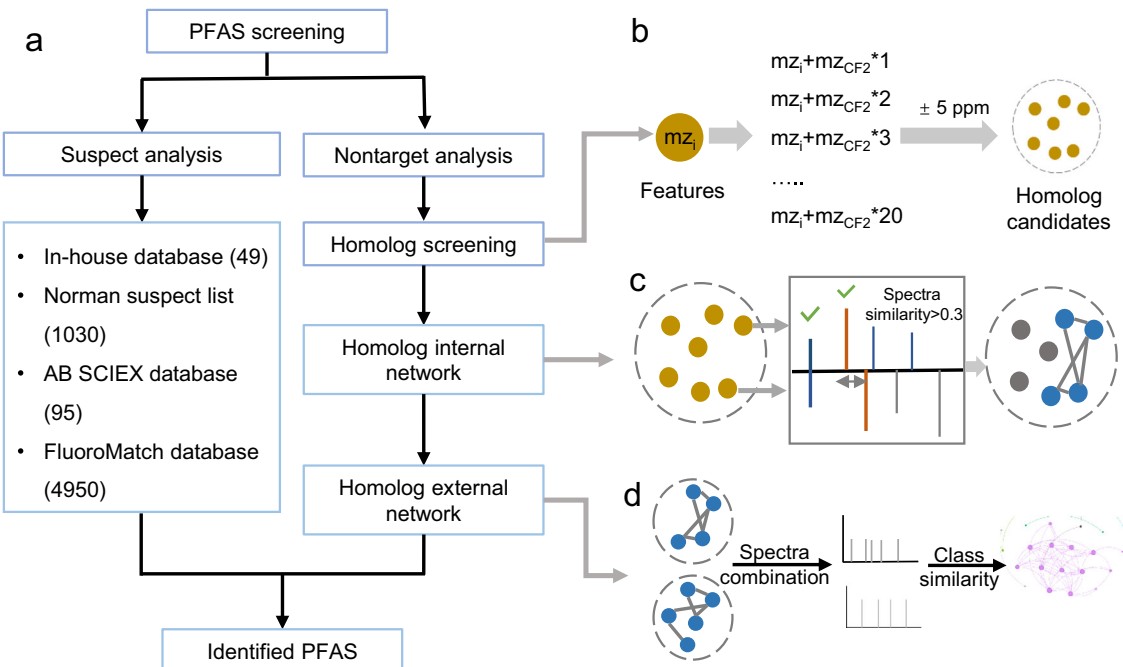

**Fig. 1 | Workflow and illustration of homolog screening and network construction. a** Workflow of nontarget and suspect analysis. **b** Illustration of homolog screening. The $mz_i$ refers to the precursor m/z of features while $mz_{CF2}$ refers to the m/z of $CF_2$ unit. The dashed circle represents a potential PFAS class, and solid brown circles within the circle represent the potential homolog features. **c** An example to build an internal homolog network to filter false positive homologs. A class containing 7 features (brown circle) were input into the internal network. Four

features (blue circle) were retained because they formed edges with other features within this class, while three features (gray circle) were filtered because they did not form any connections within this class. The blue and red fragments in the spectrum represent those matched to the m/z and those matched after calculating the neutral loss, respectively. **d** Examples of building an external homolog class network to accelerate unknown PFAS identification. Spectra of each class would be merged into a single spectrum, which served as a node to construct the external network.

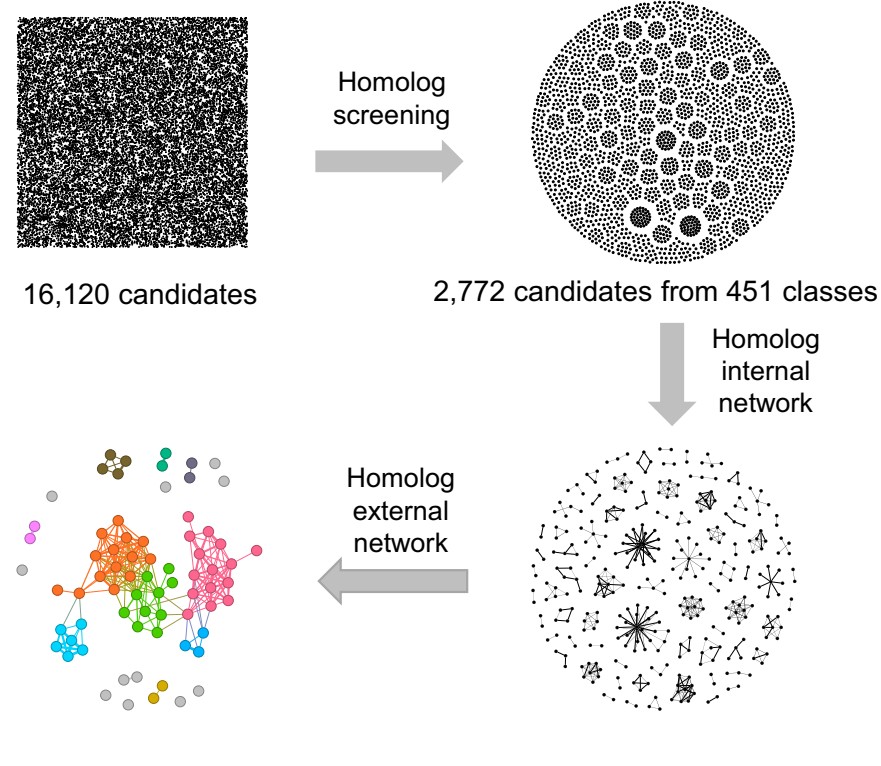

**Fig. 2 | Application of the homolog network in waterproof related samples.** Features refer to spectra peak groups, classes refer to a series of PFAS differing by $CF_2$ unit, communities refer to a group of PFAS classes clustered by class spectra similarity. The black nodes refer to features. The colored circles represent classes, while colors represent the different communities. Source data are provided as a Source Data file.

increased by 11% after using the internal network (Supplementary Fig. 1b), which may be due to the low spectra similarity caused by different instruments. We also compared our screening results with FluoroMatch 4.5[39] and FindPFΔS[25] using the standard sample. We found that our approach has fewer false positive and false negative features than FluoroMatch and significantly lower false negatives relative to FindPFΔS (Supplementary Fig. 1c).

Waterproof products, including car rubber wipers, textiles, waterproof cloth, and waterproof chemicals, along with related industrial sludge, were analyzed using liquid chromatography coupled with high-resolution mass spectrometry with an electrospray ionization source in negative ion mode. A total of 61,726 features were extracted from the mass spectrometry raw data, of which 16,120 features were assigned MS/MS spectra, through peak picking and alignment using MS-DIAL[40]. We then extracted 2772 PFAS candidates belonging to 451 classes based on the homolog screening script. A homolog internal network was constructed for each homolog class after filtering the spectra. The internal network comprised 332 nodes belonging to 71 clusters, indicating 332 PFAS candidates of 71 classes (Fig. 2). Examples of the internal network and classes that did not pass through the internal network are shown in the Supplementary Fig. 2 and 3. The spectra of each class were combined into a single spectrum, which served as a node to construct the homolog external network. The nodes of the external network were grouped into 17 communities through the Louvain community detection algorithm[38]. The results of suspect analysis were annotated in the external network.

### Identification of PFAS based on the external network

PFAS were identified with the assistance of the external network. The external network clustered structurally similar substances together for simultaneous identification. An example is shown in Fig. 3a, fragments of 134.98726 Da, 300.97479 Da, and 466.95529 Da were observed in the spectra of community a. The neutral losses of 165.98753 Da and 165.9805 Da between these fragments correspond to $C_3F_6O$ (165.98533 Da) with mass errors of +2.20 mDa and -4.84 mDa, indicating the presence of multiple ether bonds (Fig. 3a). The sequential neutral loss of $C_3F_6O$ has also been found in the spectra of chloro-perfluoropolyether carboxylates reported in in New Jersey soils[41]. Therefore, these fragments were annotated as $[C_2F_5O]^-$ (error -0.22 mDa), $[C_5F_{11}O_2]^-$ (error +1.98 mDa) and $[C_8F_{17}O_3]^-$ (error -2.86 mDa). The ultimate neutral loss of 143.98679 Da corresponds to $C_2F_4CO_2$. Finally, Node 23 has been identified as perfluorotriether carboxylic acids (PFTrECAs). We observed the same fragments of $[C_2F_5O]^-$, $[C_5F_{11}O_2]^-$ and $[C_8F_{17}O_3]^-$ and the extra fragment of $[C_{11}F_{23}O_4]^-$ in the spectra of Node 196, indicating an additional $C_3F_6O$ for Node 196 compared to Node 23. Therefore, Node 196 was identified as perfluoropentanether carboxylic acids (PFTeECAs). Similarly, Node 451 was identified as perfluoropentanether carboxylic acids (PFPeECAs).

In addition, the external network groups structurally similar suspect classes with unknown classes, enabling annotations to propagate throughout the network. A series of similar fragments were found in the spectra of community b (Fig. 3b). We identified the fragments of $[C_{11}F_{15}]^-$, $[C_{11}F_{17}]^-$, $[C_{12}F_{17}O]^-$ and $[C_{12}HF_{18}O]^-$ based on the spectra of n:2 FTOHs and n:3 FTCAs in this community. The subsequent fragment of 522.98163 Da in Node 181 was identified as $[C_{12}H_2F_{19}O]^-$. The neutral loss of 112.03623 Da corresponds to $H_2F_2$ and $C_2H_4CO_2$, indicating the presence of the carboxyl group. The fragment of 89.0231 Da was identified as $[C_3H_5O_3]^-$, indicating that the oxygen is attached to the β carbon. Finally, node 181 was identified as n:2:3 fluorotelomer ether carboxylic acids (n:2:3 FTECAs).

The external network also clusters in-source fragments and neutral losses, preventing duplicate annotations for the same PFAS.

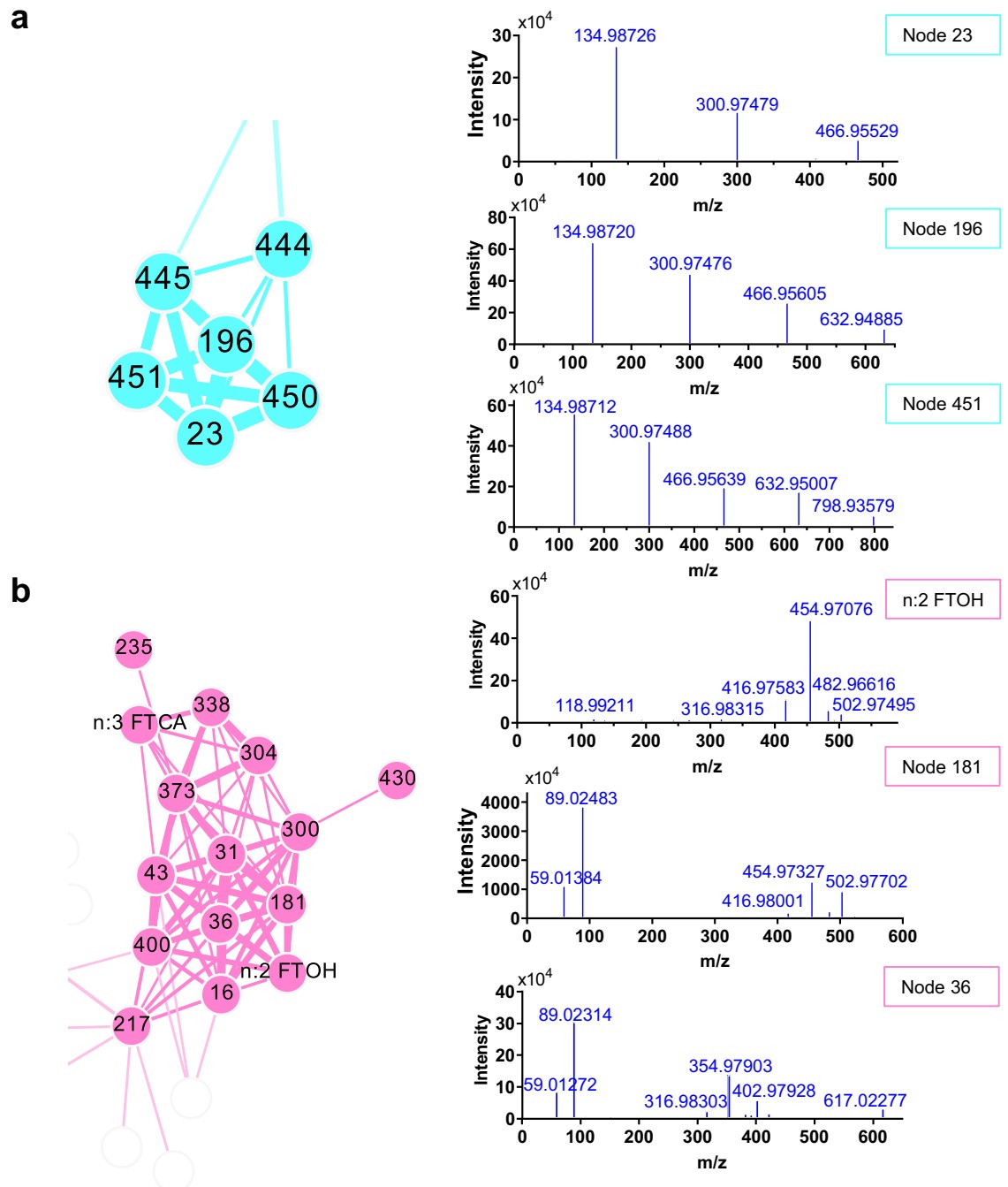

**Fig. 3 | Spectra of nodes in community a and b of external network. a** The part of community a (blue circle) in the external network and the MS/MS spectra of nodes in community a. **b** The part of community b (bright red circle) in the external network and the MS/MS spectra of nodes in community b.

The spectra of Node 36 were the same as those of Node 181 (n:2:3 FTECAs, Fig. 3b). The retention times of all the homologs of these two classes were within 0.05 min tolerance. After comparing their chromatograms, Node 36 was annotated as the $[M + C_2F_3]^-$ adduct of n:2:3 FTECAs. Other nodes of this class, except for Node 235, were also identified as in-source fragments and adducts of n:2:3 FTECAs (Supplementary Fig. 4), indicating this class undergoes complex source-internal reactions in the mass spectrometry. Similarly, Node 444 was annotated as the $[M + C_2F_3]^-$ adduct of Node 451 in the community a. Precursor m/z of Nodes 445 and 450 correspond to $[C_8F_{17}O]^-$ and $[C_{11}F_{23}O]^-$ and these two nodes were subsequently annotated as in-source fragments of Node 196 and Node 451 through the chromatograms.

Another five PFAS classes were also identified based on the external network, including n:3 fluorotelomer ether carboxylic acids (n:3 FTECAs), n:4 fluorotelomer carboxylic acids (n:4 FTCAs), double bond hydrogen substituted perfluoroalkyl carboxylic acids (dH-PFCAs), hydrogen substituted n:1 fluorotelomer alcohols (H-n:1 FTOHs) and n:2 fluorotelomer ether carboxylic acids (n:2 FTECAs). Details of the identification process of these classes are provided in Supplementary Note 1.

**Identified PFAS**

A total of 94 PFAS from 18 classes were identified after combining the results of nontarget and suspect analysis, with 16 of these confirmed through commercial standards (Supplementary Data 1). The

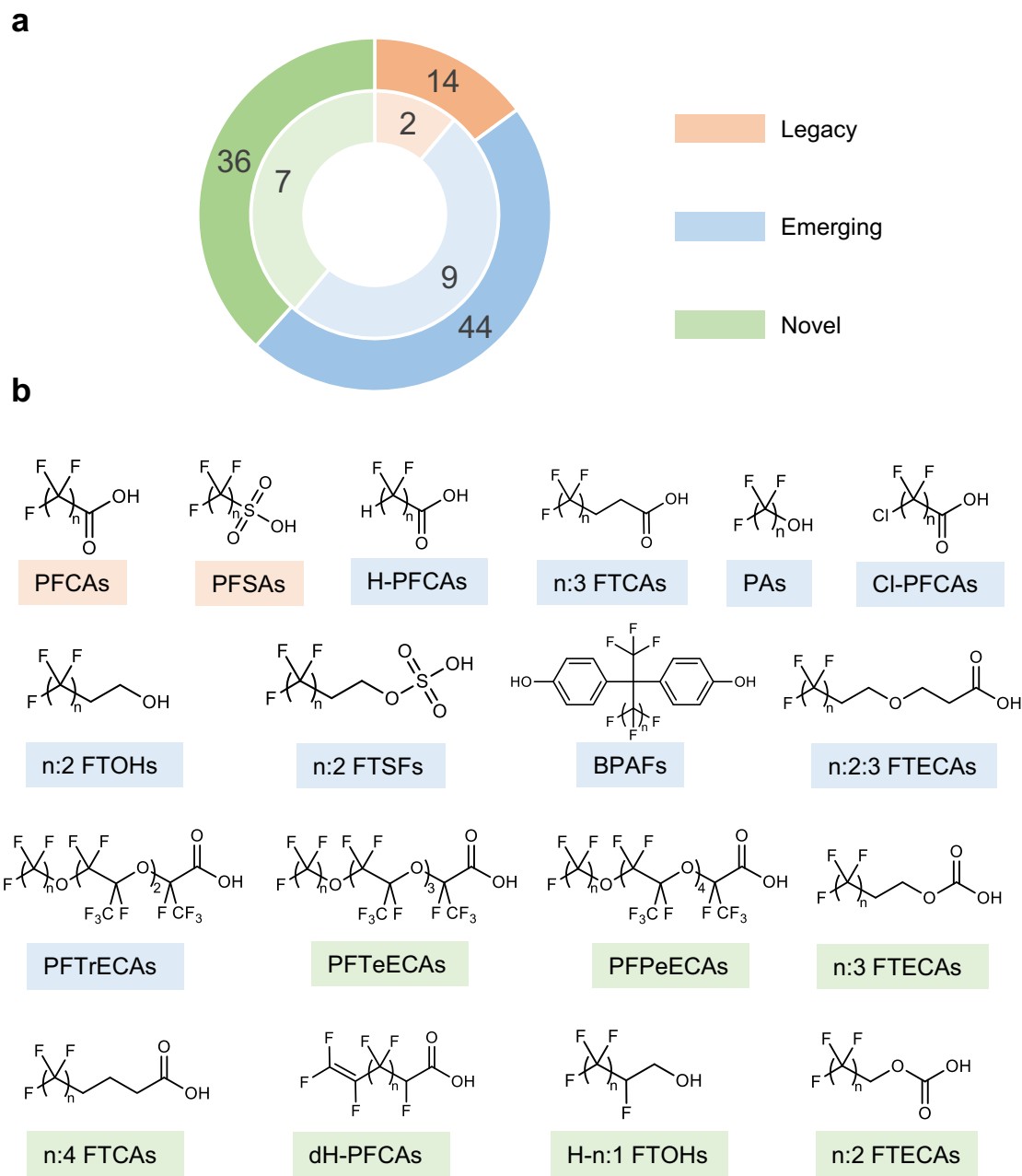

**Fig. 4 | Ninety-four identified PFAS from 18 classes detected in waterproof related samples. a** The inner ring represents the number of legacy (light orange), emerging (light blue) and novel (light green) PFAS class while the outer ring represents the number of legacy (orange), emerging (blue) and novel (green) PFAS. **b** Inferred structures of 18 identified PFAS classes. Orange represents the legacy class, blue represent the emerging class, and green represents the novel class.

diagnostic fragments and neutral losses of identified classes were list in Supplementary Table 1. PFAS are classified into legacy, emerging and novel PFAS. The legacy PFAS refer to PFCAs and PFSAs. Emerging PFAS refer to other PFAS that have been previously reported in literatures. Novel PFAS refer to PFAS that have not been previously detected in any matrix, which was determined based on the list of emerging PFAS compiled from previous literatures and reviews (Supplementary Data 2, updated in August, 2024), as well as references linked to the structure in PubChem and relevant literature retrieved through Sci-Finder. We identified 14 legacy PFAS, 44 emerging PFAS, and 36 novel PFAS, respectively (Fig. 4a, b). Spectra of emerging and novel PFAS were provided in Supplementary Fig. 5–24.

Nine emerging PFAS classes were detected in the waterproof related samples, including hydrogen-substituted perfluorocarboxylic

acids (H-PFCAs), n:3 fluorotelomer carboxylic acids (n:3 FTCAs), perfluoroalkyl alcohols (PAs), chlorine-substituted perfluorocarboxylic acids (Cl-PFCAs), n:2 fluorotelomer alcohols (n:2 FTOHs), n:2 fluorotelomer sulfate (n:2 FTSF), bisphenol AFs (BPAFs), n:2:3 FTECAs, and PFTrECAs. H-PFCAs are common emerging PFAS that have been widely detected in environmental and human samples[4,42,43]. The class of n:3 FTCAs was reported as metabolites of n:2 FTOHs[44] and has been detected in marine mammals[16] and beluga whales[42]. The class of n:2 FTOHs has been reported in the air and in samples taken around fluorochemical manufactures in Japan and can transform into PFCAs[45]. The class of n:2 FTSFs was reported as metabolites of n:2 FTOHs[46] and has been detected in the food contact materials[35]. BPAF is widely used as an alternative to BPA, leading to its frequent detection in environmental and biota samples[47,48]. However, this is the first report of the

perfluoroalkyl homolog of BPAF. C11, C13, and C15 homologs of n:2:3 FTECAs have been reported in a recent literature about food packaging paper. Although the structures have not been determined there, the proposed structures and spectra of this class are consistent with ours[35]. However, we have identified unreported C12, C14, C16, C17, and C19 homologs of this class.

Seven classes of novel PFAS were discovered, comprising PFTeE-CAs, PFPeECAs, n:3 FTECAs, n:4 FTCAs, dH-PFCAs, H-n:1 FTOHs, n:2 FTECAs. Among these, C15-PFTeECA, C18-PFPeECA, 7:3 FTECA, n:4 FTCAs, and n:2 FTECAs have been recorded in PubChem, indicating their potential commercial use. The structures of PFTeECAs and PFPeECAs are similar to Gen-X, which has been widely used as a PFOA alternative and detected around the world[49]. These classes may have same usage as Gen-X. In addition, n:4 FTCAs, n:2 FTECAs, and 7:3 FTECA have been recorded in Japan Chemical Substance Dictionary (http://jglobal.jst.go.jp/en/), suggesting that they may be used in Japan.

### Detection of PFAS in samples

We investigate the detection of PFAS in the waterproof products and sludge samples (Supplementary Fig. 25). Most PFAS were detected in the waterproof chemicals and the industrial sludge samples, which is reasonable considering that they are primary source samples. Three classes of FTOH derivatives, including H-n:1 FTOHs, n:2:3 FTECAs and n:2 FTSFs were also detected in the textile and car wiper samples, indicating the prevalent presence of FTOH compounds in the waterproof related samples. These results consist with a previous study against paper products[35], where n:2:3 FTECAs and n:2 FTSTs were also detected. PFAS detected in the waterproof chemical were markedly different from those detected in the industrial sludge. PFAS classes including n:2 FTOHs, n:4 FTCAs, dH-PFCAs and n:2 FTECAs were only detected in waterproof chemical, and therefore may be characteristic contaminants of this source. BPAFs and perfluoroalkyl polyether carboxylic acids were detected only in the industrial sludge, indicating that they were derived from the manufacture of other products.

We compared the total peak areas of each PFAS class. PFCAs exhibited the highest peak areas in downstream products, followed by PFSAs and H-PFCAs (Supplementary Table 2). In contrast, emerging and novel PFAS showed higher peak areas compared to legacy PFAS in the waterproof chemical and sludge samples. The class of n:2:3 FTECAs exhibited the highest peak areas in waterproof chemicals. This class is related to patents from Japan and could be a new ingredient added to the waterproof chemical. We may have underestimated the proportion of this substance because of in-source fragments. Classes of n:2 FTSFs and dH-PFCAs also accounted for a notable proportion in the chemical. BPAFs exhibited the highest peak areas in the industrial sludge sample. BPAF and its homolog have rarely been reported in PFAS studies. However, BPAF have been reported to have reproductive, developmental, and neurotoxic effects[50–52], posing considerable threats to human health and ecosystems. Our results indicate that fluorochemical manufacturers may be significant sources of BPAF and its homolog.

### Retrospective database mining

We searched the spectra of identified PFAS in a local dataset and a public sample database to investigate their occurrences in previous samples. The local dataset consists of three air samples collected in Tsukuba (Japan) in 2020, four rice samples cultivated using wastewater from Japan in 2016, and three wastewater samples collected from Wuhan (China) in 2005 (see "Methods" section). Fifty identified PFAS were detected in the local dataset (Fig. 5a), mainly in wastewater samples collected from Wuhan. Emerging classes, including H-PFCAs, n:3 FTCAs, Cl-PFCAs, and PAs were observed in wastewater, air, and rice. These PFAS were also reported in the wastewater from a fluorochemical manufacturing park in Changshu[53]. The class of n:2:3 FTECAs was detected in the wastewater samples collected from 2005 while

firstly reported in a literature published this year[35]. Our results suggested that this class may have been in production and use for decades. Five novel classes, including n:4 FTCAs, dH-PFCAs, H-n:1 FTOHs, n:2 FTECAs, and n:3 FTECAs were also detected in the local dataset. These novel PFAS were predominantly detected in the wastewater samples, indicating that they have existed for a long time before discovered. Novel PFAS of dH-PFOA were also detected in the air samples from Japan (Fig. 5a), indicating that they have been released into the environment.

Meanwhile, we searched the spectra of the identified PFAS in the public database MassIVE to investigate their occurrences in previous samples using MASST[54] (see "Method" section). A total of 28 PFAS were found in the MassIVE databases through the batch MASST search algorithm (Fig. 5b). The mirror plots of these matched PFAS are provided in Supplementary Fig. 26–53, and the work links of these matching are available in Supplementary Data 3. Emerging PFAS, including H-PFCAs, n:3 FTCAs, 6:2 FTSF, and BPAF, have been found in the environment, biota, and even human samples from 15 countries (Fig. 5c). These PFAS have been detected in air, wastewater, and mammals previously[16,42,43,47]. Here, we indicated their widespread distribution all over the world. We also found 6:4 FTCA, 10:4 FTCA, H-6:1 FTOH, and dH-PFOA in previous samples, although we reported these PFAS for the first time. Among these PFAS, 6:4 FTCA was found in human breast milk samples collected from the United States. H-6:1 FTOH, dH-PFOA, and 6:4 FTCA were also detected in local dataset, indicating the consistence of the results between local dataset and public database. In conclusion, these PFAS have preexisted in the environment and human body.

## Discussion

The emergence of novel PFAS presents significant challenges for global chemical management. Advanced technologies are required to promptly detect and evaluate the presence and impact of these compounds, given their potential adverse effects on both the environment and human health[55,56]. In this study, we proposed a two-layer networking approach based on the fragmentation pathway of PFAS. This approach builds upon homolog screening by integrating intraclass molecular network, which is the first layer internal network, reducing 94% of the false positive candidates. This approach further constructs the external network based on the class spectra similarity. The external network enables simultaneous identification of structurally similar classes, deduction from known to unknown classes, and the aggregation of in-source fragments and adducts.

Utilizing this approach, we identified 36 novel PFAS in waterproof-related samples, highlighting previously unreported PFAS in waterproof products. The identification of novel PFAS is the foundation for further studies on their environmental occurrence and toxicology. We constructed a local dataset and developed a batch MASST search algorithm for the retrospective analysis of identified PFAS. Fifteen novel PFAS were found in the local dataset, and four were found in the public database. The retrospective analysis of PFAS reveals the potential global exposure of emerging and novel PFAS.

Although our work has identified dozens of novel PFAS, the synthesis of PFAS standards is required in future studies to verify the structures of novel PFAS. Extensive efforts are needed for further toxicity evaluations of these discovered PFAS and those identified in previous literatures[21,57]. Managing and controlling persistent chemicals is a formidable task that requires collaborative efforts and comprehensive policies. The limited sample size used in this study may not fully capture the release of PFAS associated with the important application of waterproofing. Furthermore, PFAS analysis critically depends on MS/MS information, and future investigations could leverage iterative data-dependent acquisition or data-independent acquisition methodologies to improve MS/MS coverage and enhance the

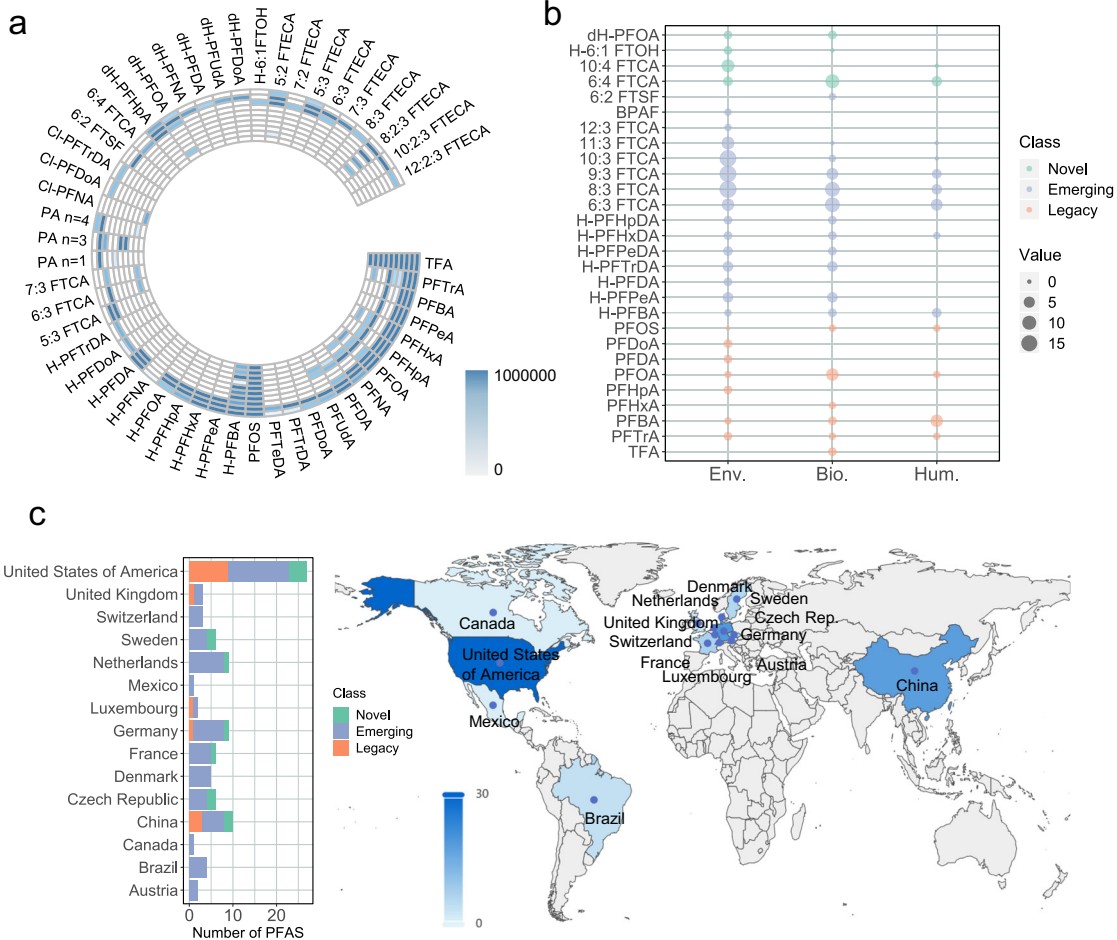

**Fig. 5 | Detection and distribution of identified PFAS in the local dataset and public database. a** Detection of identified PFAS in local dataset, the shade of the color represents the peak area. From the inner ring to the outer ring, the columns represent: three air samples, four rice samples, and three wastewater samples, respectively. The heatmap was generated using the circlize package in R. **b** Identified PFAS in public MassIVE database, the value refers to the number of the datasets in which each PFAS was found. PFAS were matched at confidence level 3, indicating that only the presence of functional groups is confirmed, and there may be matches with isomers. Env., Bio., and Hum. refer to environmental, biota, and human samples, respectively. The bubble diagram was generated using the ggplot2 package in R. **c** The distribution of identified PFAS in the public database. Darker color indicates a higher number of datasets. The bar chart was generated using the package ggplot2 in R and the world map was generated using the package pyecharts in python. Source data are provided as a Source Data file.

robustness of the analysis. Lastly, this study employed the MassIVE database and self-built dataset for retrospective data mining, but the low proportion of environmental samples in the MassIVE database may result in inadequate representation of actual environmental pollution. MASST is unable to distinguish between PFAS isomers with same fragmentation patterns, suggesting that the matched results may correspond to structurally similar PFAS. As the matching is conducted on raw data files, the matched PFAS may originate from background blanks rather than the actual samples. However, PFAS in the blanks also underscores their potential widespread presence in the environment. This study confirms that large-scale mass spectrometry data can provide new perspectives for monitoring and analysis of chemicals, and efforts should be made to promote the establishment of public environmental databases.

## Methods
### Chemicals
HPLC grade methanol (MeOH) was purchased from Merck (Germany). Ammonium acetate (HPLC grade) was purchased from CNW Technologies GmbH (Germany). Ethyl acetate (EtOAc, 99.5%; pesticide residue-PCB analytical grade) was supplied by FUJIFILM Wako Pure Chemical Corporation (Osaka, Japan). Water (LCMS, purity> 99.99%)

was purchased from Thermo Fisher (U.S.A.). Information about PFAS standards is provided in Supplementary Data 4.

### Sample Information and Extraction
Four types of consumer products, including rubber car wipers ($n = 4$), textile ($n = 5$), waterproof cloth ($n = 1$), waterproof chemical ($n = 2$), were obtained from a market in Osaka, Japan in January 2022. Two sludge sample were collected from the wastewater drainage of a large fluorochemical industry in Osaka, Japan in January 2022 for comparison with consumer products. The rubber wipers of car were obtained by purchasing car accessories. Textile samples refer to carpets while waterproof cloth refer to clothes with waterproof properties. Waterproof chemicals refer to waterproofing spray liquids. The brand information for these products and the manufacture information for sludge samples were not provided due to commercial confidentiality concerns.

The rubber wiper, textile, and waterproof cloth were cut into 5 mm pieces and mixed well before extraction. A further extraction procedure was applied to 5 g of solid products, 1 ml of waterproof chemical and 50 g (wet weight) of industrial sludge. The sample was extracted by adding 10 ml of methanol and 10 ml of ethyl acetate and then shaken for 30 min using an orbital shaker. Extract was transferred

into 50 ml polypropylene tube and centrifuged at 1247 g for 10 min. Subsequently, supernatant was passed through carbon SPE (Supel-clean™Envi-Carb™, Sigma-Aldrich Corp., St. Louis, MO, USA;100 mg; 1 mL) followed by three washings with 1 mL methanol. The extract was concentrated under nitrogen, replaced with methanol, adjusted into 1 ml and used for instrumental analysis.

## Instrument analysis
Instrument analysis was conducted using ultra-high performance liquid chromatography (UPLC; UltiMate 3000 Series; Thermo Fisher Scientific) tandem quadrupole orbitrap mass spectrometry (Q Exactive Focus; Thermo Fisher Scientific). The liquid chromatography separation was performed on a BEH C18 column (ACQUITY UPLC BEH C18, 1.7 μm, 2.1 × 150 mm, Waters, USA) at 40 °C. The mobile phase was water containing 2 mM ammonium acetate and methanol and the gradient elution is shown in Supplementary Table 3. The mass spectrometry data were collected in negative electrospray ionization mode (ESI-), with mass ranges of 80–1000 Da and 50–1000 Da for full scans under mass resolution 70,000 (3 Hz) and data dependent MS/MS scans under mass resolution 17,500 (12 Hz), respectively. The aux gas heated temperature and capillary temperature were set to 413 °C and 320 °C, respectively. The spray voltage and S-lens RF level were set to −2500 V and 50, respectively. The sheath gas, aux gas and spare gas were set to 48, 11, 2, respectively. The normalized collision energy was set to −35 ± 15 eV. The automatic gain control (AGC) target was set on 1e6 and 5e4 for full scan and MS/MS scan, and precursor isolation window was set at 1.0 Da. The dynamic exclusion was applied to capture more diverse MS/MS spectra, with an exclusion time set to 6 s. This allows for each ion to be selected only once for MS/MS analysis within each 6-second interval.

## Nontarget and suspect analysis
The nontarget method was developed based on molecular network enhanced homolog screening. In details, peaks with intensities greater than 10000 were extracted from raw data using MS-DIAL ver. 4.70[40] and then filtered by $CF_2$ normalized mass defect (0–0.15 and 0.85–1)[21,36]. Homolog screening was conducted for features with MS/MS spectra based on the precursor ion mass difference of 49.99681 Da ($CF_2$)[36,53]. The MS1 tolerance and the minimum homologs per class for homolog screening were set at 5 ppm and 2, respectively. We then filtered the MS/MS spectra of these potential PFAS homologs to improve the quality of the network. Fragments in MS/MS spectra were retained based on the following rules: (1) m/z > 60 Da; (2) relative intensity > 5 %; and (3) mass difference with the precursor >1 Da. Subsequently, a two-layer homolog network was constructed using Gephi 0.10.1. The first layer is the homolog internal network, constructed for each potential homolog class separately (Fig. 1c). The node within the internal network exhibited one homolog in the class, whereas the edge of the internal network represented the spectra similarity between nodes. The spectra similarity was calculated using the GNPS spectra similarity algorithm[37], which is modified cosine similarity algorithm that considers neutral losses, where the MS/MS tolerance is set to 0.01 Da. Edges with a similarity greater than 0.3 were retained in the internal network. The parameter of 0.3 was assessed based on false positive and false negative rates (Supplementary Fig. 54). Features were removed if they did not have any edge connecting them to other features in the same class. The edge labels represent common fragments between two nodes. Common fragments refer to those with the same mass or differing by $CF_2$ units. The internal network was used to filter false homologs in potential classes and provide common fragment information of classes to assist in structure annotation.

After filtering using internal network, the spectra of each class were merged into a single spectrum (Fig. 1d). Only common fragments

between homologs, which serve as edge labels in the internal network, are included in the merged spectrum. These merged spectra were further used to construct the second layer homolog network, the external network based on the spectra similarity which was calculated using the following Eq. (1):

$$S = \frac{F \times 2}{(F_1 + F_2)} \tag{1}$$

where $S$ refers to the similarity of two spectra, $F$ refers to the number of matched fragments in these two spectra with tolerance of 0.005 Da, and $F_1$ and $F_2$ refer to the number of total fragments in the two spectra. Edges with $S$ greater than 0.2 would be retained. Thus, the homolog external network reflected the distance of structure among homolog classes. Finally, community detection was performed on the homolog external network using the Louvain algorithm[38], implemented through the community detection module in Gephi. This analysis was conducted with a resolution setting of 1.0 and considered the edge weights, which is class spectra similarity ($S$). PFAS were structurally diagnosed according to their communities. The molecular formulae of precursors, fragments, and neutral losses of candidate PFAS were calculated using a python script. The maximum elemental composition was set as follows: C:30, H:30, F:30, O: 6, N: 2, P: 2, S: 2. Cl and Br were included depending on the isotope pattern. The mass error was set to 5 ppm for precursors and 0.01 Da for fragments and neutral losses, while the ring and double bond number range was set from -0.5 to 10. The structures were inferred based on the masses of the precursors, fragments, and neutral losses.

Suspect screening was performed using a list consisting of an in-house standard database, the Norman suspect list[18], AB SCIEX database, and the FluoroMatch database[39] through MS-DIAL. The MS1 tolerance was set at 0.002 Da.

Finally, the identified PFAS were assigned a 1–5 confidence level according to the system proposed by Schymanski et al. [58]. If the spectra similarity between two classes exceeds 0.5 and their retention times are consistent within tolerance of 0.05 min, the chromatograms of these two classes would be manually checked to identify the molecular ions, in-source fragments and adducts[59]. The peak areas of the identified PFAS were calculated using Tracefinder 4.1.

## Retrospective analysis
The local dataset consists of three air samples, four rice samples, and three wastewater samples. The three air samples were collected using a portable active air sampler, collecting both particle phase and gas phase (using polyurethane foams and activated charcoal as adsorbent), and the sampling lasted for around 48 h at a constant flow rate of 20 L min$^{-1}$, resulting in an average total volume of 53 m$^3$[60]. The four rice samples were initially cultivated in the laboratory using industrial wastewater for one day before seedling transplantation. After this initial period, they were grown using tap water until harvest[61]. Three wastewater samples were collected in three industries from Wuhan in 2005, and samples were treated by solid phase extraction[62]. These samples were stored at -80°C after extraction and subsequently reinjected under the same instrument conditions as those for waterproof products. The exact mass, retention time and spectra of identified PFAS were fed into MS-DIAL to search their occurrence in local MS database.

MassIVE is a comprehensive repository that contains 9,393,372 mass spectrometry files from 15,620 datasets uploaded by researchers worldwide. These 15,620 datasets come from nearly 50 countries, among which 7719 datasets are annotated through National Center for Biotechnology Information (NCBI) classification. This includes 3,569 human datasets and 149 environmental datasets, covering samples such as air, lakes, wastewater, and sludge. These datasets and related information are available for download as well as for online browsing

(https://massive.ucsd.edu/ProteoSAFe/datasets.jsp#%7B%22query%22%3A%7B%7D%2C%22table_sort_history%22%3A%22createdMillis_dsc%22%7D). The MASST search tool returns the origin of the matched MS/MS spectrum with respect to the dataset and file information and any metadata associated with the file.

For our final analysis, we utilized the following MASST search settings: a single spectrum search was conducted via the online GNPS workflow (https://ccms-ucsd.github.io/GNPSDocumentation/), with data filtered to exclude all MS/MS fragment ions within ±17 Da of the precursor m/z. MS/MS spectra were window-filtered to retain only the top 6 fragment ions within a ± 50 Da window across the spectrum. The precursor ion mass tolerance was set to 0.005 Da, and the MS/MS fragment ion tolerance was set to 0.01 Da. Library spectra were filtered similarly to the input data. Matches between input spectra and library spectra required a cosine similarity score above 0.6 and at least 2 matched peaks (1 matched peak was set for C2-C4 PFCAs and n:2 FTSFs considering that only one fragment exists in the spectra). The mass accuracy setting and MS/MS spectrum matching thresholds were validated using PFAS standards (35 PFAS), a menthol blank sample (2207 spectra), and a local spectra database (830 PFAS and 16,734 non-PFAS) as provided in Supplementary Note 3 and Supplementary Method. Searching the 35 PFAS standards and 830 local PFAS spectra against the local non-PFAS database, the false discovery rate is 0 and 0.017 under our search settings, which is acceptable.

However, MS/MS is often unable to distinguish structurally similar PFAS isomers with similar structures. Therefore, the MASST query results are defined as level 3, which means functional groups can be identified. Finally, the matched datasets were classified into environmental, biota and human samples manually based on their species information and descriptions.

### Quality assurance/quality control (QA/QC)

Twenty-one PFAS standards, including C4-C14, C16 and C18 perfluoroalkyl carboxylic acids (PFCAs), C6-C10, C12 perfluoroalkane sulfonic acids (PFSAs) were used to evaluate the recoveries, relative standard deviation (RSD), and sensitivity of the instrument. The recoveries and relative standard deviation were evaluated by spiking 20 ng standard to Florisil for each PFAS, and treated as waterproof samples. The recoveries of PFAS ranged from 74.3% to 107.7%, with RSD below 15% (Supplementary Table 4). Instrument sensitivity and mass accuracy were evaluated by injecting standards at 1, 5, 10, 50 and 100 µg/L. All PFAS standards could be detected at 1 µg/L and the precursor mass errors were within ±0.002 Da. A procedure blank and a solvent blank were injected in each batch of analysis. Only PFOS was detected in the solvent and procedure blank, with a peak area approximately one-tenth of that observed for the lowest injected standard (1 µg/L). Blank correction was performed by subtracting three times the PFOS peak area in the blank from the corresponding peak areas in the samples.

### Reporting summary

Further information on research design is available in the Nature Portfolio Reporting Summary linked to this article.

## Data availability

The data that support the findings of this study are available within the article and its Supplementary Information files. The raw mass spectrometry files of the samples and the local dataset generated in this study are available in the MassIVE database under the accession number MSV000095719 [https://doi.org/10.25345/C5251FX35]. The analysis data generated in this study are available in Zenodo 13626193. [https://zenodo.org/records/13626193]. Source data are provided with this paper.

## Code availability

The code used in this study are available in Zenodo 13482320.

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

## Acknowledgements

This study was supported by the National Key Research and Development Program of China (2023YFC3706601, S.W.), the National Natural Science Foundation of China (22206075, X.W.), the Strategic Priority Research Program of the Chinese Academy of Sciences (XDB0750100, S.W.), KAKENHI (15H02587, 21H04949, N. Yamashita), KAKENHI (18H03394, 20KK0245, S.T.), and the Environment Research and Technology Development Fund (JPMEERF20211G02, N. Yamashita and JPMEERF20245001, S.T.) of the Environmental Restoration and Conservation Agency of Japan provided by Ministry of the Environment of Japan.

## Author contributions

S.W. convinced and supervised the study, Z.J., and S.T. wrote the manuscript, Z.J., N. Yu, and X.W. carried out the data analysis, S.T. and N. Yamashita collected samples and performed the experiments.

## Competing interests

The authors declare no competing interests.
