## [Transparent Peer Review file · Nature Communications]

Two-layer homolog network approach for PFAS nontarget screening and retrospective data mining

Corresponding Author: Dr Si Wei

Version 0:

Reviewer comments:

Reviewer #1

(Remarks to the Author)

The paper describes the application of a data evaluation to detect and identify PFAS in HRMS data and presents some interesting finding.

In general, the presented method and the results are not sufficiently described for a scientific publication. There is missing a clear presentation and explanation of the approach, the results and the discussion. For example, the authors used Kendrick mass analysis combined with diagnostic fragments to prioritize potential PFAS from other signals in the HRMS data and apply the network clustering to structure the data. This is not explicitly explained and discussed compared to similar approaches already described in the literature. There is also missing information on the identification approach used; in this case I only see suspect screening which depends on compound and MS information in libraries. Therefore, it is not quite clear how real "novel PFAS" could be identified.

There are missing also details in the measurement procedure, e.g. to acquire MS-MS spectra, details on samples and the databases (local vs. global; which information is available in these data bases? Or is it a collection of HRMS measurement data).

In conclusion, I cannot recommend publication of the paper. Due to some interesting results on the occurrence of specific PFAS, I would recommend re-submission of a considerably revised version of the paper.

Further comments:

There is missing information and context, so that the text of the abstract can be comprehended.

I. 17: Measurement by LC-neg. ESI-HRMS should be added.

Explain, what are interferential candidates?

I. 19: What are the 5 distinct PFAS communities? How are they characterized?

I. 20: In which samples did you find the novel PFAS? Are these PFAS novel in the samples investigated, or generally not yet described (but how did you identify these PFAS then?)

I. 21: What is a local database?

I. 23: Why should the PFAS have been used for decades when you detected them only once in one specific sample?

Explain which public database?

I. Define "novel PFAS"! Novel in which context?

I. 48-51: KMD is not used as the only criterion for data prioritization. Give more specific examples from the literature! There are some helpful recent review paper on PFAS screening!

I. 58: Explain "case studies"! What is the message you want to tell here?

I. 62-68: Give more details on environmental samples available in MASST and which HRMS data can be obtained (MS-MS data? Etc.).

I. 86-88: It is obscure from which samples and measurements you are talking about.

I. 89: Which criteria did you use to select the homologs and how did you define the 291 classes?

I. 91-93: Give more information on the criteria for the internal network. Is it a subdivision of the homologs based on diagnostic fragments or MS-MS fragmentation patterns?

I. 97 – 99: Give more information on the external network (procedure, criteria).

I. 107-109: This class is based on diagnostic fragments on ether compounds. Did you select them manually or is it a result of the clustering process?

I. 111-112: This is a manual extraction of exact mass; which m/z at which mass accuracy?

I. 112-116: Explain, how did you distinguish in-source fragments from molecular ions?

- l. 120 -124: Fig. 1 is too small to be readable.
 - l. 134-136: Are you really sure, that all the novel PFAS haven't been reported in the literature?
 - l. 170-172: Give more details and info on the samples!
 - l. 196-197: Give more info on the local database. This point is completely obscure! How many samples from which sites are in the data base; which MS information is available etc.?
 - l. 206: Also more info and details are required for the global samples in MASST.
 - l. 211-214: Which human species data have been used here?
 - l. 282-286: Info on MS-MS data acquisition are missing!
 - l. 295-300: Give more information and parameters on network construction. Also, some references would be helpful, if modifications of published methods were used.
 - l. 300, 309: Fig. numbers should be Fig. 1, isn't it?
- Further information on NTS workflow on compound identification, the local and global data base are needed.

Reviewer #2

(Remarks to the Author)

The manuscript by Jiao et al combined MS1 based homolog screening and MS2 based molecular networking to identify novel PFAS chemicals from commercial products. They noticed that the molecular networking step greatly reduced the false positive rates compared with homolog screening alone. They detected 96 PFASs in the commercial products, 46 PFASs reported for the first time, and screened these novel PFASs in local environmental samples as well as public metabolomics datasets on the MassIVE database. They revealed that although these PFASs were first reported in literature, they already exist in samples collected in the past.

The manuscript holds novelty in the combined application of homolog screening and molecular networking, and the search of novel PFASs in public datasets are good applications. My major concerns are (1) the lack of false negative evaluation for the molecular networking step (thus over-estimated efficacy in false positive removal), (2) the false positives in public database query based on MS2 only (especially for PFASs), (3) the necessity for the "external network", (4) and the lack of information on data and code availability.

Major comments:

1. Line 86-95, Line 299-307: There is not a clear documentation on how the molecular networks were constructed neither in the Methods or the Results sections. Is the "internal network" built for each homolog series separately? What are the network parameters you used? Why there are 291 "classes" after the homolog screening, but only 82 classes after the molecular networking? Is it because no MS2 spectra were collected for the other classes, or because their MS2 spectra lack similarity? When you claim that the internal network filtered 85% of false positive candidates, how did you confirm that it did not lead to false negative? What are some examples of "classes" that pass homolog screening but does not pass the molecular network screening? In Figure S1, what are the labels on the network edges, why sometimes there are multiple labels on the same edge?
2. Line 96-118: While I appreciate the visualization from this "external network", I find it not producing scientific insights. This paragraph is mostly descriptive. I suggest the authors to remove this analysis, or reduce the amount of descriptive texts and focus on the scientific implication from this "external network". Also, I do not see a clear need for generating a unified spectrum for each PFASs class. Is it possible to just use the MS2 spectra of each PFASs to build a large, unified network?
3. I am very concerned about the low thresholds used for MASST queries especially for PFASs. Please justify your choice of using 2 matching peaks and 0.5 cosine similarity. Previous research has shown that MS2 fragments will not be able to distinguish the numerous PFASs isomers. The level of structural confidence in Figure 3 needs to be clearly communicated. Could you test the FDR of your search using your local databases?
4. I did not find any information on code and data availability. I suggest the authors to make their analysis, raw data, and reference spectra publicly available.

Minor comments:

1. Line 20-24, Line 80: In the abstract mentioning types and numbers of samples you used for PFASs discovery and detection. Same applies to the end of Introduction.
2. Line 98: It is unclear how the network was divided to 10 communities. Be specific about your parameters.
3. The texts in Figure 1b-d and the sample names in Figure 3a and 3c are not visible.
4. Figure 2a: what is the inner ring showing?
5. Line 208: How did you classify the public datasets into these categories? Through manual inspection?
6. Line 262: How many products were sampled in each category?
7. Line 297: Please clarify criteria three here. Did you get rid of fragments from a spectrum if the fragments are present in blank samples? If yes I do not understand the reason to do this filtering.

Version 1:

Reviewer comments:

Reviewer #2

(Remarks to the Author)

The author has adequately addressed my comments. I appreciate the author's effort to test the workflow parameters in detail and include results in the Supplementary Information.

(Remarks on code availability)

Reviewer #3

(Remarks to the Author)

The manuscript entitled “Global and Historical Use of Novel Contaminants Revealed by Two-Layer Homolog Network and Database Mining” was thoroughly revised by the authors. It now has a clearer method description, the important parameters from the network analysis are provided and the identification procedures are sufficiently described. Nevertheless, I still see some important issues concerning the transparency and interpretation of the results, especially the robustness of the findings of novel PFAS in the online database, that need to be addressed before publication.

Major comments:

- The current presentation of the detection of newly identified PFAS in the online database (MassIVE) is questionable! As I understood, the acquired MS2 spectra of newly identified PFAS were matched with MS2 spectra (from HRMS raw data) in the database. The authors used an MS1 tolerance of 5 mDa, 2 (!) matched peaks, and a modified cosine similarity > 0.5. I am highly concerned that the rate of false positives can be high with those parameters. Many PFAS have rather similar MS2 spectra although their structures are different, and they become even more similar when using the “modified cosine similarity” which considers neutral losses and mass shifts. There are clear examples needed that show spectral similarity (mirror plot) between the database spectra and the ones identified in this study! If hits are only on identification level 3, then they might not be the newly identified PFAS but other structures. Furthermore, since HRMS raw data files are present in the database, the MS2 spectra could also come from background signals not originating from the samples. A clearer description of assumptions and limitations is needed!

- I could not find any information about your MS2 coverage. Was iterative data-dependent acquisition with multiple injections used to get more MS2 spectra? It is stated that 16000 features were detected, but I assume not all of them have MS2 information. As I understood, the network analysis always needs MS2 information. How were features considered that do not have MS2 spectra? Please explain.

- In general, it is a good idea to make a comparison between existing approaches and your new approach. However, given the very limited amount of data, also here limitations need to be discussed. The comparison between Fluoromatch, FindPFAS, and the author’s approach is only based on a single sample containing 32 PFAS. Those software tools are highly dependent on the chosen parameters and the authors only applied the default parameters without trying to find suitable ones for their data. I would suggest moving some details of the discussion of this comparison into the SI.

Furthermore, the authors state that it is difficult to determine the number of non-PFAS in a PFAS standard. Why is that difficult? Do you consider also background compounds in this kind of analysis? If yes, then no reliable FPR rate can be calculated.

- Please consider also checking PubChem (besides literature) to verify that the structures you have identified are indeed novel. It is difficult these days (with so many publications on PFAS) to check every single PFAS NTS paper and the chance of overlooking novel compounds in literature is high.

Additional minor comment:

- Abstract: It is hard to understand the approach used for PFAS NTS from the abstract, especially the reduction of false positives by 94%. Please consider explaining this in more detail.

- In my opinion the word “local database” is misleading. If I understood it correctly, this data comes from a sample collection of only 10 samples which is not a database. Please consider changing the wording.

- I noticed several small typos and grammatical mistakes in the manuscript that must be corrected.

- I think in the introduction there are still some PFAS prioritization approaches missed that are important. There are other successful approaches besides KMD analysis and diagnostic fragments (e.g., CCS vs. m/z plots or m/C ratios) that can also drastically reduce false positives. Those methods can have even advantages over mass differences since they do not rely on any homologs. In lines 48 – 58, it sounds like other methods cannot identify novel PFAS. Please consider a slight rewriting.

- Line 49: “resulting in high false positive rates...”.

- Line 65: Shouldn’t it be “MassIVE”?

- Line 78: There are many studies on PFAS in waterproof chemicals, especially textiles! Please cite the existing literature properly.

- Line 127: I would not call that purified, since only signals above a fixed threshold were considered as noise and removed.

- Line 142: Isn’t this mass error rather high given the specified resolution of the Orbitrap?

- Line 233-239: How was it assessed which classes are dominant? All data is qualitative. Were peak areas used? Please consider how different ionization efficiencies in ESI can be. This needs to be clarified!

- Line 290: Waterproof products are a well-known source for PFAS and there are many studies investigating this. Please rewrite the sentence.

- QA/QC section: This section covers only the most basic quality measures. In my opinion, if the PFOS background really reaches close to 1 µg/L is very high and concerning. How was the blank correction performed, I could not find a proper explanation. This needs a more detailed discussion!

- Line 496: Several times “Zendo” was written instead of “Zenodo”.

(Remarks on code availability)

Version 2:

Reviewer comments:

Reviewer #3

(Remarks to the Author)

I checked the author's revisions on my suggestions. The authors have thoroughly addressed the comments and clarified the identified issues. I appreciate the effort, in particular, the provision of MS/MS spectra matches from the large MS database which improves the manuscript's transparency and might help other researchers find those novel PFAS.

(Remarks on code availability)

Disposition of Reviewers' comments

REVIEWER COMMENTS

Reviewer #1 (Remarks to the Author):

The paper describes the application of a data evaluation to detect and identify PFAS in HRMS data and presents some interesting finding.

In general, the presented method and the results are not sufficiently described for a scientific publication. There is missing a clear presentation and explanation of the approach, the results and the discussion. For example, the authors used Kendrick mass analysis combined with diagnostic fragments to prioritize potential PFAS from other signals in the HRMS data and apply the network clustering to structure the data. This is not explicitly explained and discussed compared to similar approaches already described in the literature. There is also missing information on the identification approach used; in this case I only see suspect screening which depends on compound and MS information in libraries. Therefore, it is not quite clear how real “novel PFAS” could be identified.

Response: Thank you for your valuable suggestions. We have made the following modifications based on your suggestions:

(1) We have provided a detailed supplement to the methods and related references. “We developed a two-layer homolog network (Fig. 1a) for the nontarget analysis of PFAS, with specific details provided in the Methods section. Briefly, precursor ions differing by CF_2 units were selected as candidates through homolog screening (Fig. 1b)³³. After spectra purification, the first-layer internal network was constructed for each candidate class based on GNPS spectra similarity (similarity > 0.3, Fig. 1c)³⁴. The spectra of each class were then merged into a single spectrum, and the second-layer external network was constructed based on the class spectra similarity (similarity > 0.2, Fig. 1d). The nodes in the external network were grouped into communities based on the Louvain community detection algorithm³⁵, with the resolution parameter set to 1.0.” *(Line 95-103)*.

We added references and parameters for the homolog screening, “Homolog screening was conducted for features with MS/MS spectra based on the precursor ion mass difference of

49.99681 Da (CF₂) as previous studies^{33,51}. The MS1 tolerance and the minimum homologs per class for homolog screening were set at 5 ppm and 2, respectively.” (Line 361-364) Details about the internal and external network were provide in the Methods section, “The first layer is the homolog internal network, constructed for each potential homolog class separately (Fig. 1c). The node within the internal network exhibited one homolog in the class, whereas the edge of the internal network represented the spectra similarity between nodes. The spectra similarity was calculated using the GNPS spectra similarity algorithm³⁴, which is modified cosine similarity algorithm that considers neutral losses, where the MS/MS tolerance is set to 0.01 Da. Edges with a similarity greater than 0.3 were retained in the internal network. The parameter of 0.3 was assessed based on false positive and false negative rates (Supplementary Fig. 26). Features were removed if they did not have any edge connecting them to other features in the same class. The edge labels represent common fragments between two nodes. Common fragments refer to those with the same mass or differing by CF₂ units. The internal network was used to filter false homologs in potential classes and provide common fragment information of classes to assist in structure annotation.” (Line 368-381) “Edges with S greater than 0.2 would be retained. Thus, the homolog external network reflected the distance of structure among homolog classes. Finally, community detection was performed on the homolog external network using the Louvain algorithm³⁵, implemented through the community detection module in Gephi. This analysis was conducted with a resolution setting of 1.0 and considered the edge weights, which is the class spectra similarity (S).” (Line 391-396)

(2) We assessed the false positives and false negatives of the method based on our local mass spectrometry database (which includes 133 PFAS and 16,734 non-PFAS) and in-house standards. We compared our method with conventional homolog screening, Fluoromatch, and FindPFAS. The results indicate that our method has lower false positive and false negative rates compared to previous methods. The relevant methods and results have been included in the main text and the supplementary information. “Our internal network which integrates homolog screening and molecular network while utilizing both MS1 and MS/MS information, could reduce false positives in homolog screening. We evaluated the denoising capability of the internal network using the local spectra database and an in-house standard sample. Utilizing the

local spectra database, we observed that compared to homolog screening, the internal homolog network demonstrated a 94% reduction in false positive features (Supplementary Fig. 1a). The false negative rate increased by 11% after using the internal network (Supplementary Fig. 1b), which may be due to the low spectra similarity caused by different instruments. We also compared our screening results with FluoroMatch 4.5³⁶ and FindPFAS²³ using the standard sample. We found that our approach has fewer false positive and false negative features (Supplementary Fig. 1c) than FluoroMatch. This is reasonable considering that FluoroMatch is based on the homolog screening. Our approach demonstrates comparable false positives and significantly lower false negatives relative to FindPFAS²³, which employs MS/MS spectra information for prioritization. This evaluation suggests that our approach effectively integrates MS1 and MS/MS information, thereby leveraging the strengths of current strategies.” (Line 104-119) “We evaluated the denoising capability of the internal network using a local mass spectra database and an in-house standard sample (100 µg/L). The local spectra database contains 133 PFAS from 45 classes, while the standard samples include 32 PFAS from 6 classes. MS1 tolerance was set at 5 ppm for homolog screening and internal network (Supplementary Fig. 1a). Methanol blanks were used in FluoroMatch³⁶ for filtering and the full-scan peak intensity threshold was set at 10,000, the same as the threshold we used in peak picking. The precursor mass tolerance was set to 0.002 Da, and the MS/MS fragment tolerance was set to 0.01 Da for FluoroMatch. We utilized the "fragment differences" module of FindPFAS for comparative analysis. The fragment difference settings were configured as follows: FindPFAS_1: CF₂, FindPFAS_2: CF₂, C₂F₄, HF, and FindPFAS_3: CF₂, C₂F₄, HF, CF₃, CF₂O. The parameters for these configurations were set to: Number of Differences Desired: 1.0, Fragment Mass Tolerance: 0.01 Da, Intensity Threshold: 5.0, Remove Multiple Mass Tolerance: 0.002, and Occurrence Number Threshold: 20.0. Additionally, we evaluated the analytical method using typical fragments with the “diagnostic fragment” module of FindPFAS. For this evaluation, the fragments were set to the default values, including [CF₃]-, [PO₂F]-, [SO₂F]-, [C₂F₅]-, [C₃F₇]-, [C₄F₉]-, [C₅F₁₁]-, [C₆F₁₃]-, and [C₇F₁₅]-.

False positive features refer to non-PFAS that were incorrectly identified as PFAS, and false negative features refer to PFAS that were not identified as PFAS. The false positive rate (FPR) was calculated by dividing the number of false positive features by the number of non-PFAS in

the database. The false negative rate (FNR) was calculated by dividing the number of false negative features by the number of PFAS in the database. The total number of non-PFAS in the standard samples is difficult to determine. Therefore, the FPR and FNR were not calculated for the standard sample.” (Line 415-438)

Supplementary Figure 1. The comparison of homolog network with other methods. **a** The number of false positive features and false positive rates of homolog screening (HS) and internal network (IN) using the local mass spectrometry database. HS_2 and HS_3 refers to HS with a minimum of 2 homologs per class and 3 homologs per class, respectively. IN_2 and IN_3 refers to IN with a minimum of 2 homologs per class and 3 homologs per class, respectively. **b** The number of false negative features and false negative rates of homolog screening (HS) and internal network (IN) using 100 ug/L PFAS standard sample. **c** The number of false positive features and false negative features of internal network, FluoroMatch, FindPFAS_1 with CF₂, FindPFAS_2 with CF₂, C₂F₄, and HF, and FindPFAS_3 with CF₂, C₂F₄, HF, CF₃, and CF₂O, the diagnostic fragment module of FindPFAS.

(3) we have added the information on the identification process in the manuscript.

“PFAS were identified with the assistance of the external network. The external network clustered structurally similar substances together for simultaneous identification. An example was shown in Figure 3a, fragments of 134.98726 Da, 300.97479 Da and 466.95529 Da were observed in the spectra of community a. The neutral losses of 165.98753 Da and 165.9805 Da between these fragments corresponds to C₃F₆O (165.98533 Da), indicating the presence of multiple ether bonds (Fig. 3a). The sequential neutral loss of C₃F₆O has also been found in the spectra of chloro-perfluoropolyether carboxylates reported in in New Jersey soils³⁸. Therefore, these fragments were annotated as [C₂F₅O]-, [C₅F₁₁O₂]- and [C₈F₁₇O₃]-. The ultimate neutral loss of 143.98679 Da corresponds to C₂F₄CO₂. Finally, Node 23 has been identified as perfluorotriether carboxylic acids (PFT₃ECAs). We observed the same fragments of [C₂F₅O]-, [C₅F₁₁O₂]- and [C₈F₁₇O₃]- and the extra fragment of [C₁₁F₂₃O₄]- in the spectra of Node 196, indicating an additional C₃F₆O for Node 196 compared to Node 23. Therefore, Node 196 was identified as perfluoropentanether carboxylic acids (PFT₅ECAs). Similarly, Node 451 was identified as perfluoropentanether carboxylic acids (PFPeECAs).

In addition, the external network groups structurally similar suspect classes with unknown classes, enabling annotations to propagate throughout the network. A series of similar fragments were found in the spectra of community b (Fig. 3b). We identified the fragments of [C₁₁F₁₅]-, [C₁₁F₁₇]-, [C₁₂F₁₇O]- and [C₁₂HF₁₈O]- based on the spectra of n:2 FTOHs and n:3 FTCAs in this community. The subsequent fragment of 522.98163 Da in Node 181 were identified as [C₁₂H₂F₁₉O]-. The neutral loss of 112.03623 Da corresponds to H₂F₂ and C₂H₄CO₂, indicating the presence of the carboxyl group. The fragment of 89.0231 Da corresponds to [C₃H₅O₃]-, indicating that the oxygen is attached to the β carbon. Finally, node 181 was identified as n:2:3 fluorotelomer ether carboxylic acids (n:2:3 FTECAs).

The external network also clusters in-source fragments and neutral losses, preventing duplicate annotations for the same PFAS. The spectra of Node 36 were the same as those of Node 181 (n:2:3 FTECAs, Fig. 3b). The retention time of all the homologs of these two classes consist within 0.05 min tolerance. After comparing their chromatograms, Node 36 was annotated as the [M+C₂F₃]- adduct of n:2:3 FTECAs. Other nodes of this class except for Node 235 were also identified as in-source fragments and adducts of n:2:3 FTECAs (Supplementary Fig. 4), indicating this class undergoes complex source-internal reactions in the mass

spectrometry. Similarly, Node 444 were annotated as $[M+C_2F_3]^-$ adduct of Node 451 in the community a. Precursor m/z of Nodes 445 and 450 correspond to $[C_8F_{17}O]^-$ and $[C_{11}F_{23}O]^-$ and these two nodes were subsequently annotated as in-source fragments of Node 196 and Node 451 through the chromatograms.

Another five PFAS classes were also identified based on the external network, including n:3 fluorotelomer ether carboxylic acids (n:3 FTECAs), n:4 fluorotelomer carboxylic acids (n:4 FTCAs), double bond hydrogen substituted perfluoroalkyl carboxylic acids (dH-PFCAs), hydrogen substituted n:1 fluorotelomer alcohols (H-n:1 FTOHs) and n:2 fluorotelomer ether carboxylic acids (n:2 FTECAs). The identification process of these classes was provided in Supplementary Note 1.” *(Line 137-180)*

Figure 3. Spectra of nodes in community a and b of external network. **a** The part of community a (blue circle) in the external network and MS/MS spectra of nodes in community a. **b** The part of community b (bright red circle) in the external network and MS/MS spectra of nodes in community b.

There are missing also details in the measurement procedure, e.g. to acquire MS-MS spectra, details on samples and the databases (local vs. global; which information is available in these data bases? Or is it a collection of HRMS measurement data).

Response: Thank you for your feedback. We have added information on measurement procedure, samples, and databases.

(1) Here are the added information on measurement procedures in Methods section, “The mass spectrometry data were collected in negative electrospray ionization mode (ESI-), with mass ranges of 80–1000 Da and 50–1000 Da for full scans under mass resolution 70,000 and data dependent MS/MS scans under mass resolution 17,500, respectively. The aux gas heated temperature and capillary temperature were set to 413 °C and 320 °C, respectively. The spray voltage and S-lens RF level were set to -2500 V and 50, respectively. The sheath gas, aux gas and spare gas were set to 48, 11, 2, respectively. The normalized collision energy was set to -35 ± 15 eV. The automatic gain control (AGC) target was set on $1e6$ and $5e4$ for full scan and MS/MS scan, and precursor isolation window was set at 1.0 Da.” (Line 347-356)

(2) Added details about the samples, “Four types of consumer products, including rubber wiper of car (n=4), textile (n=5), waterproof cloth (n=1), waterproof chemical (n=2), were obtained from a market in Osaka, Japan in January 2022. Two sludge sample was collected from the wastewater drainage of a large fluorochemical industry in Osaka, Japan in January 2022 for comparison with consumer products. The rubber wipers of car were obtained by purchasing car accessories. Textile samples refer to carpets while waterproof cloth refer to clothes with waterproof properties. Waterproof chemicals refer to waterproofing spray liquids. The brand information for these products and the manufacture information for sludge samples were not provided due to commercial confidentiality concerns.” (Line 318-327)

(3) Added details on the local database, “The local database consists of three air samples, four rice samples, and three wastewater samples. The three air samples were collected using a portable active air sampler, collecting both particle phase and gas phase (using polyurethane foams and activated charcoal as adsorbent), and the sampling lasted for around 48 h at a constant flow rate of 20 L min^{-1} , resulting in an average total volume of 53 m^3 . Detailed sample pretreatment methods for air samples can be found in one of our previous studies⁵⁸. The four rice samples were initially cultivated in the laboratory using industrial wastewater for one day before seedling transplantation. After this initial period, they were grown using tap water until harvest. For a detailed description of the sample pretreatment methods, please refer to our previous study⁵⁹. Three wastewater samples were collected in three industries from Wuhan in

2005, and samples were treated by solid phase extraction as one of our previous studies⁶⁰.”
(Line 440-451)

(4) The public database MassIVE hosts an expansive collection of high-resolution mass spectrometry (HR-MS) data contributed by researchers worldwide. Currently, it encompasses 9,393,372 mass spectrometry files from 15,620 datasets. This extensive repository features a diverse array of environmental and biological samples, each accompanied by detailed sample and instrumental information provided by the contributing researchers. The MassIVE database allows users to download datasets and browse online. The MASST search tool within MassIVE facilitates finding matched MS/MS spectra and provides the origin of these spectra, including detailed dataset and file information, as well as any associated metadata. We have incorporated this into the main text: “MassIVE is a comprehensive repository that contains 9,393,372 mass spectrometry files from 15,620 datasets uploaded by researchers worldwide. These 15,620 datasets come from nearly 50 countries, among which 7,719 datasets are annotated through National Center for Biotechnology Information (NCBI) classification. This includes 3,569 human datasets and 149 environmental datasets, covering samples such as air, lakes, wastewater, and sludge. These datasets and related information are available for download as well as for online browsing (https://massive.ucsd.edu/ProteoSAFe/datasets.jsp#%7B%22query%22%3A%7B%7D%2C%22table_sort_history%22%3A%22createdMillis_dsc%22%7D%22%7D). The MASST search tool returns the origin of the matched MS/MS spectrum with respect to the dataset and file information and any metadata associated with the file.” *(Line 455-465)*.

In conclusion, I cannot recommend publication of the paper. Due to some interesting results on the occurrence of specific PFAS, I would recommend re-submission of a considerably revised version of the paper.

Response: We extend our sincere gratitude to you for your time and dedication. Your insightful comments and recommendations have been crucial in enhancing the quality of this manuscript. In response, we added the description of the method's details and parameters, compared our method with homolog screening, FluoroMatch and FindPFAS. Additionally, we have

supplemented information regarding the identification process of novel PFAS, sample information, mass spectrometry acquisition methods, as well as information about the local database and the public database MassIVE. We hope these revisions address your concerns.

Further comments:

There is missing information and context, so that the text of the abstract can be comprehended.

Response: Thank you for your suggestions. We have added the necessary information and revised the abstract to enhance its comprehensibility, as shown below.

I. 17: Measurement by LC-neg. ESI-HRMS should be added.

Response: Thank you for your suggestions. Due to the abstract's word limit of 150 words, we have included a general reference to mass spectrometry in the revised abstract. Specifically, we have stated: “Here, we developed a two-layer homolog network approach for PFAS nontarget screening using mass spectrometry.” *(Line 16-17)*

Explain, what are interferential candidates?

Response: Thank you for your comments. Interferential candidates refer to features whose precursor ion masses comply with homologs rules but are not actually PFAS homologs. These features would be screened out with PFASs by homolog screening and interfere with identification. We have revised the sentence to clarify, “The first layer incorporates molecular network on homolog screening, reducing false positives by 94%.” *(Line 17-18)* An example of the interferential candidates were provided in Supplementary Figure 3.

Supplementary Figure 3. Examples of classes that pass the homolog screening but do not pass internal network. **a** Four spectra of the class m/z 441.018868, 491.01419, 591.00897, 691.00128 Da; **b** Two spectra of the class m/z 610.98816, 790.9967 Da.

I. 19: What are the 5 distinct PFAS communities? How are they characterized?

Response: Thank you for your comments. A network community refers to a subset of nodes within a network where the connections between nodes in the same subset are denser, while the connections with nodes in other communities are sparser. Community algorithms analyze the network structure to automatically identify these natural divisions. These 5 distinct communities were characterized through the Louvain community detection algorithm. We have revised this sentence, “The second layer expediting the identification by categorizing PFAS classes through class spectra similarity.” (*Line 18-19*) The details about community was provided in the main text rather than abstract, “Finally, community detection was performed on the homolog external network using the Louvain algorithm³⁵, implemented through the community detection module in Gephi. This analysis was conducted with a resolution setting of 1.0 and considered the edge weights, which is the class spectra similarity (S).” (*Line 393-396*)

I. 20: In which samples did you find the novel PFAS? Are these PFAS novel in the samples

investigated, or generally not yet described (but how did you identify these PFAS then?)

Response: Thank you for your comments. We found novel PFAS in the industrial sludge sample and waterproof products including the rubber wiper of a car, textiles, waterproof cloth, and waterproof chemicals. These novel PFAS have not been reported in any research papers. We have revised this sentence to clarify, “A total of 94 PFAS were detected in twelve waterproof products and two related industrial sludge, including 37 novel PFAS not previously reported in any literature.” *(Line 19-21)*. Our identification process involved using an internal homolog network to screen PFAS classes from thousands of peak features, followed by clustering PFAS classes using an external network. We then deduced the structures of the unknown PFAS based on their fragmentation patterns and known PFAS classes. We have also provided additional details on the identification process in the results section as mentioned in the major comments *(Line 137-180)*.

I. 21: What is a local database?

Response: The local database, as opposed to a public database, is constructed by re-injecting the samples we previously collected into the mass spectrometry. The local database is constructed for retrospective analysis. We have revised this sentence to make it clear, “A local database was constructed for retrospective analysis by re-injecting our previous samples, revealing fifteen novel PFAS in samples collected in 2005.” *(Line 21-23)*

I. 23: Why should the PFAS have been used for decades when you detected them only once in one specific sample?

Response: Thank you for your comments. These novel PFAS were detected in samples from 2005 in the local database as well as in the waterproof related samples of this study. Four of them were also detected in the MassIVE database, indicating that they might have emerged early and globally. Our statement was not inaccurate enough considering these PFAS may be the transformation products, so we have removed this sentence. Additionally, as we mentioned in the introduction, reports on novel PFAS typically lag far behind their production and emissions. For instance, 6:2 chlorine substituted perfluoroalkyl ether sulfonic acid was used for 30 years before being reported. P-perfluorous nonenoxybenzenesulfonate has been

manufactured since the 1980s and has been reported recently.

Explain which public database?

Response: The public database refers to MassIVE database constructed by Global Natural Products Social (GNPS). We have revised this sentence to clarify, “The retrieval of the public database MassIVE uncovered the detection of novel PFAS in samples from seven countries.” (Line 23-24)

1. Define “novel PFAS”! Novel in which context?

Response: Novel PFAS refers to PFAS that have not been detected in any matrix. We have revised the sentence to clarify, “A total of 94 PFAS were detected in twelve waterproof products and two related industrial sludge, including 37 novel PFAS not previously reported in any literature.” (Line 19-21) The definition of novel PFAS was provided in the results section, “Novel PFAS refer to PFAS that were not previously reported in any matrix, which was determined based on the list of emerging PFAS compiled from previous literatures and reviews (Supplementary Data 2, updated in August, 2024).” (Line 187-190)

1. 48-51: KMD is not used as the only criterion for data prioritization. Give more specific examples from the literature! There are some helpful recent review paper on PFAS screening!

Response: Thank you for pointing this out. We have reviewed literatures on PFAS screening and supplemented it in the introduction, “Mass defect, Kendrick mass defect analysis, and chemical formula assignment are commonly used nontarget screening methods for PFAS¹⁹⁻²¹. However, these methods only consider the mass of the precursor ions, resulting in high false positive in complex environment samples. Diagnostic fragments, fragment differences, neutral losses, and fragment ion flagging are also used for prioritizing PFAS based on MS/MS spectra^{22, 23}. Nevertheless, these features in MS/MS spectra are summarized from limit and known PFAS, which hinders the discovery of novel PFAS.” (Line 47-53)

1. 58: Explain “case studies”! What is the message you want to tell here?

22table_sort_history%22%3A%22createdMillis_dsc%22%7D). The MASST search tool returns the origin of the matched MS/MS spectrum with respect to the dataset and file information and any metadata associated with the file.” (Line 455-465)

I. 86-88: It is obscure from which samples and measurements you are talking about.

Response: Thank you for your comment. We have revised this sentence to clarify, “Waterproof products including the rubber wiper of a car, textiles, waterproof cloth, and waterproof chemicals, along with related industrial sludge, were analyzed using liquid chromatography coupled with high-resolution mass spectrometry with an electrospray ionization source in negative ion mode.” (Line 120-123)

I. 89: Which criteria did you use to select the homologs and how did you define the 291 classes?

Response: Thank you for your comment. We selected PFAS homologs by precursor ion mass differences of 49.99681 Da (CF₂). If two features differ by CF₂ units, they are classified into the same class. These 291 classes were screened out by the homolog screening. We have supplemented the screening criteria and relevant references here, “Briefly, precursor ions differing by CF₂ units were selected as candidates through homolog screening (Fig. 1b)³³.” (Line 96-98) “Homolog screening was conducted for features with MS/MS spectra based on the precursor ion mass difference of 49.99681 Da (CF₂) as previous studies^{33, 51}. The MS1 tolerance and the minimum homologs per class for homolog screening were set at 5 ppm and 2, respectively.” (Line 361-364)

I. 91-93: Give more information on the criteria for the internal network. Is it a subdivision of the homologs based on diagnostic fragments or MS-MS fragmentation patterns?

Response: Thank you for your comment. The internal network was constructed for each class, where nodes represent homologs within the class, and edges represent spectral similarity between homologs. It could be considered as a subdivision based on MS-MS fragmentation patterns. We have added the criteria to clarify, “The first layer is the homolog internal network, constructed for each potential homolog class separately (Fig. 1c). The node within the internal

network exhibited one homolog in the class, whereas the edge of the internal network represented the spectra similarity between nodes. The spectra similarity was calculated using the GNPS spectra similarity algorithm³⁴, which is modified cosine similarity algorithm that considers neutral losses, where the MS/MS tolerance is set to 0.01 Da. Edges with a similarity greater than 0.3 were retained in the internal network. The parameter of 0.3 was assessed based on false positive and false negative rates (Supplementary Fig. 26). Features were removed if they did not have any edge connecting them to other features in the same class. The edge labels represent common fragments between two nodes. Common fragments refer to those with the same mass or differing by CF₂ units. The internal network was used to filter false homologs in potential classes and provide common fragment information of classes to assist in structure annotation.” (Line 368-381)

1. 97 – 99: Give more information on the external network (procedure, criteria).

Response: Thank you for your comment. The external network was constructed based on the class spectra similarity. The nodes represent PFAS classes while the edges represent class spectra similarity between nodes. Edges with similarity greater than 0.2 would be retained. We have added the information here, and in the Methods section, “The spectra of each class were then merged into a single spectrum, and the second-layer external network was constructed based on the class spectra similarity (similarity > 0.2, Fig. 1d). The nodes in the external network were grouped into communities based on the Louvain community detection algorithm³⁵, with the resolution parameter set to 1.0.” (Line 99-103) “After filtering using internal network, the spectra of each class were merged into a single spectrum (Fig. 1d). Only common fragments between homologs, which serve as edge labels in the internal network, are included in the merged spectrum. These merged spectra were further netted to form the second layer homolog network, the external network based on the spectra similarity which was calculated using the following equation:

$$S = F \times 2 / (F1 + F2) \quad (1)$$

where S refers to the similarity of two spectra, F refers to the number of matched fragments in these two spectra with tolerance of 0.005 Da, and $F1$ and $F2$ refer to the number of total fragments in the two spectra. Edges with S greater than 0.2 would be retained. Thus, the homolog external network reflected the distance of structure among homolog classes. Finally,

community detection was performed on the homolog external network using the Louvain algorithm³⁵, implemented through the community detection module in Gephi. This analysis was conducted with a resolution setting of 1.0 and considered the edge weights, which is the class spectra similarity (*S*).” (Line 382-396)

l. 107-109: This class is based on diagnostic fragments on ether compounds. Did you select them manually or is it a result of the clustering process?

Response: Thank you for your comment. These two classes are polyether compounds with many common fragments. They were clustered together in the same community through external network. It is a result of the clustering process and community detection. We have rewritten this part, “PFAS were identified with the assistance of the external network. The external network clustered structurally similar substances together for simultaneous identification. An example was shown in Figure 3a, fragments of 134.98726 Da, 300.97479 Da and 466.95529 Da were observed in the spectra of community a. The neutral losses of 165.98753 Da and 165.9805 Da between these fragments corresponds to C₃F₆O (165.98533 Da), indicating the presence of multiple ether bonds (Fig. 3a).” (Line 137-142)

l. 111-112: This is a manual extraction of exact mass; which m/z at which mass accuracy?

Response: Thank you for your comment. We found perfluorotetraether carboxylic acid and perfluoropentaether carboxylic acid. Therefore, we extracted the exact mass of 776.9269 Da and 826.9237 Da for perfluorotetraether carboxylic acids (C₁₄HF₂₇O₆ and C₁₅HF₂₉O₆), with a relative mass error margin of 5 ppm. Our method has been modified and this sentence has been removed. We have optimized the homolog screening part, changing the requirement from at least 3 homologs per class to at least 2 homologs per class, considering that some PFAS classes have only two homologs. Therefore, these three classes were selected through homolog screening, and this statement has been removed.

l. 112-116: Explain, how did you distinguish in-source fragments from molecular ions?

Response: Thank you for your comment. In-source fragments were distinguished based on the spectra similarity and retention times as previous studies. We have added the criteria and

examples, “If the spectra similarity between two classes exceeds 0.5 and their retention times are consistent within tolerance of 0.05 minutes, The chromatograms of these two classes would be manually checked to identify the molecular ions, in-source fragments and adducts⁵⁷.” (Line 409-412) “The external network also clusters in-source fragments and neutral losses, preventing duplicate annotations for the same PFAS. The spectra of Node 36 were the same as those of Node 181 (n:2:3 FTECAs, Fig. 3b). The retention time of all the homologs of these two classes consist within 0.05 min tolerance. After comparing their chromatograms, Node 36 was annotated as the [M+C₂F₃]- adduct of n:2:3 FTECAs.” (Line 163-167)

I. 120 -124: Fig. 1 is too small to be readable.

Response: Thank you for your comment. We have divided Figure 1 into Figure 1 and Figure 2, and made modifications to Figure 1.

Figure 1. **a** Workflow of nontarget and suspect analysis. **b** Illustration of homolog screening. The mz_i refers to the precursor m/z of features while mz_{CF_2} refers to the m/z of CF_2 unit. **c** An example to build an internal homolog network to filter false positive homologues. A class containing 7 features (brown circle) were input into the internal network. Four features (blue circle) were retained because they formed edges with other features within this class, while three features (grey circle) were filtered because they did not form any connections within this class. **d** Examples of building an external homologue class network to accelerate unknown PFAS identification. Spectra of each class would be merged into a single spectrum, which

served as a node to construct the external network.

Figure 2. Application of the homolog network in waterproof related samples. Features refer to spectra peak groups, classes refer to a series of PFAS differing by CF_2 unit, communities refer to a group of PFAS classes clustered by class spectra similarity. Colours suggested the different communities.

l. 134-136: Are you really sure, that all the novel PFAS haven't been reported in the literature?

Response: Thank you for your comment. We have reviewed recent and past articles on PFAS discoveries, compiled a list of reported PFAS, and considered those not previously reported as novel PFAS. We have added this in the main text, “Novel PFAS refer to PFAS that were not previously reported in any matrix, which was determined based on the list of emerging PFAS compiled from previous literatures and reviews (Supplementary Data 2, updated in August, 2024).” (Line 187-190)

l. 170-172: Give more details and info on the samples!

Response: Thank you for your comment. We have added the types, number, and description on the samples here, “Four types of consumer products, including rubber wiper of car (n=4), textile (n=5), waterproof cloth (n=1), waterproof chemical (n=2), were obtained from a market in

Osaka, Japan in January 2022. Two sludge sample was collected from the wastewater drainage of a large fluorochemical industry in Osaka, Japan in January 2022 for comparison with consumer products. The rubber wipers of car were obtained by purchasing car accessories. Textile samples refer to carpets while waterproof cloth refer to clothes with waterproof properties. Waterproof chemicals refer to waterproofing spray liquids. The brand information for these products and the manufacture information for sludge samples were not provided due to commercial confidentiality concerns.” (Line 318-327)

I. 196-197: Give more info on the local database. This point is completely obscure! How many samples from which sites are in the data base; which MS information is available etc.?

Response: Thank you for your comment. We have added the types, number, sampling method and related references of samples in the local database. We have added the information to clarify, “The local database consists of three air samples collected from Tsukuba (Japan) in 2020, four rice samples cultivated using wastewater from Japan in 2016, and three wastewater samples collected from Wuhan (China) in 2005 (see Methods section).” (Line 247-250) “The local database consists of three air samples, four rice samples, and three wastewater samples. The three air samples were collected using a portable active air sampler, collecting both particle phase and gas phase (using polyurethane foams and activated charcoal as adsorbent), and the sampling lasted for around 48 h at a constant flow rate of 20 L min⁻¹, resulting in an average total volume of 53 m³. Detailed sample pretreatment methods for air samples can be found in one of our previous studies⁵⁸. The four rice samples were initially cultivated in the laboratory using industrial wastewater for one day before seedling transplantation. After this initial period, they were grown using tap water until harvest. For a detailed description of the sample pretreatment methods, please refer to our previous study⁵⁹. Three wastewater samples were collected in three industries from Wuhan in 2005, and samples were treated by solid phase extraction as one of our previous studies⁶⁰.” (Line 440-451)

I. 206: Also more info and details are required for the global samples in MASST.

Response: Thank you for your comment. The “global samples” here refer to the datasets in

MassIVE, originating from nearly 50 countries. We have changed the statement to clarify, “A total of 29 PFAS were detected in the MassIVE databases through the batch MASST search algorithm (Fig. 5b).” (Line 265-266). We have added more information for the MassIVE database in the methods section, “MassIVE is a comprehensive repository that contains 9,393,372 mass spectrometry files from 15,620 datasets uploaded by researchers worldwide. These 15,620 datasets come from nearly 50 countries, among which 7,719 datasets are annotated through National Center for Biotechnology Information (NCBI) classification. This includes 3,569 human datasets and 149 environmental datasets, covering samples such as air, lakes, wastewater, and sludge. These datasets and related information are available for download as well as for online browsing (https://massive.ucsd.edu/ProteoSAFe/datasets.jsp#%7B%22query%22%3A%7B%7D%2C%22table_sort_history%22%3A%22createdMillis_dsc%22%7D%22%7D). The MASST search tool returns the origin of the matched MS/MS spectrum with respect to the dataset and file information and any metadata associated with the file.” (Line 455-465)

I. 211-214: Which human species data have been used here?

Response: Thank you for your comment. “Human species” was a classification label in the “Species classification” information of MassIVE. The four datasets mentioned here include breast milk samples, throat swabs/nasal swabs/skin samples, dental samples, and human plasma samples. We have revised this sentence to make it simple and clear, “Among these PFAS, 6:4 FTCA was detected in human breast milk samples collected from the United State.” (Line 272-273)

I. 282-286: Info on MS-MS data acquisition are missing!

Response: Thank you for your comment. We have added the instrument parameters on MS-MS data acquisition, “The mass spectrometry data were collected in negative electrospray ionization mode (ESI-), with mass ranges of 80–1000 Da and 50–1000 Da for full scans under mass resolution 70,000 and data dependent MS/MS scans under mass resolution 17,500, respectively. The aux gas heated temperature and capillary temperature were set to 413 °C and 320 °C, respectively. The spray voltage and S-lens RF level were set to -2500 V and 50,

respectively. The sheath gas, aux gas and spare gas were set to 48, 11, 2, respectively. The normalized collision energy was set to -35 ± 15 eV. The automatic gain control (AGC) target was set on $1e6$ and $5e4$ for full scan and MS/MS scan, and precursor isolation window was set at 1.0 Da.” (Line 347-356)

I. 295-300: Give more information and parameters on network construction. Also, some references would be helpful, if modifications of published methods were used.

Response: Thank you for your comment. We have added references to homolog screening and the GNPS spectra similarity algorithm. Our approach builds upon homolog screening by adding molecular network based on the GNPS spectral similarity algorithm for each class (internal network), thereby reducing false positives in homologous screening. Additionally, we developed a method for merging class spectra, using the merged spectra as nodes to construct an external network. The external network enables clustering of structurally similar class, deduction from known to unknown class, and aggregation of in-source fragments. We have added the details and parameters on network construction, “The first layer is the homolog internal network, constructed for each potential homolog class separately (Fig. 1c). The node within the internal network exhibited one homolog in the class, whereas the edge of the internal network represented the spectra similarity between nodes. The spectra similarity was calculated using the GNPS spectra similarity algorithm³⁴, which is modified cosine similarity algorithm that considers neutral losses, where the MS/MS tolerance is set to 0.01 Da. Edges with a similarity greater than 0.3 were retained in the internal network. The parameter of 0.3 was assessed based on false positive and false negative rates (Supplementary Fig. 26). Features were removed if they did not have any edge connecting them to other features in the same class. The edge labels represent common fragments between two nodes. Common fragments refer to those with the same mass or differing by CF_2 units. The internal network was used to filter false homologs in potential classes and provide common fragment information of classes to assist in structure annotation.” (Line 368-381)

I. 300, 309: Fig. numbers should be Fig. 1, isn't it?

Response: Thank you for pointing this out. We have revised this section and checked the

numbering of other figures.

Further information on NTS workflow on compound identification, the local and global data base are needed.

Response: Thank you for your suggestions. We have added the identification process based on the external network (*Line 137-180*). We also supplemented the sample composition, sampling, and references information of the local database, and included descriptions and parameter information of the MassIVE database, as mentioned in the previous comment.

Reviewer #2 (Remarks to the Author):

The manuscript by Jiao et al combined MS1 based homolog screening and MS2 based molecular networking to identify novel PFAS chemicals from commercial products. They noticed that the molecular networking step greatly reduced the false positive rates compared with homolog screening alone. They detected 96 PFASs in the commercial products, 46 PFASs reported for the first time, and screened these novel PFASs in local environmental samples as well as public metabolomics datasets on the MassIVE database. They revealed that although these PFASs were first reported in literature, they already exist in samples collected in the past.

The manuscript holds novelty in the combined application of homolog screening and molecular networking, and the search of novel PFASs in public datasets are good applications. My major concerns are (1) the lack of false negative evaluation for the molecular networking step (thus over-estimated efficacy in false positive removal), (2) the false positives in public database query based on MS2 only (especially for PFASs), (3) the necessity for the “external network”, (4) and the lack of information on data and code availability.

Response: We sincerely thank you for your time and effort. Your professional opinions and valuable suggestions have greatly contributed to improving the quality of this manuscript. In the following, we address your suggestions point by point. Briefly, (1) we have evaluated the false negative for the molecular networking step through local spectra database and an in-house

standard sample. There are 133 perfluoroalkyl substances and 16,734 non-perfluoro substances in the database. Our method reduces the false positive features (confirmed by the structures) from 6994 to 399 while the false negative features increased from 1 to 16. (See major comments 1) (2) We have tested the performance of MASST search under different parameters using PFAS standards samples and the local spectra database. Under precursor mass tolerance to 0.005 Da and MS/MS fragment ion tolerance to 0.01 Da, the false discovery rate is 0 and 0.017 for standard samples and spectra database, respectively (See major comments 3). (3) The external network mainly serves three functions: clustering structurally similar unknown classes for simultaneous identification, deriving unknown classes from suspect classes, and clustering in-source fragments and adducts to reduce workload. We have provided the identification process and then clarify the necessity for the external network. (*Line 137-180*) (4) The raw data have been uploaded to MassIVE (MSV000095719) and code have been uploaded to Zendo (<https://zenodo.org/records/13482320>).

Major comments:

1. Line 86-95, Line 299-307: There is not a clear documentation on how the molecular networks were constructed neither in the Methods or the Results sections. Is the “internal network” built for each homolog series separately? What are the network parameters you used? Why there are 291 “classes” after the homolog screening, but only 82 classes after the molecular networking? Is it because no MS2 spectra were collected for the other classes, or because their MS2 spectra lack similarity? When you claim that the internal network filtered 85% of false positive candidates, how did you confirm that it did not lead to false negative? What are some examples of “classes” that pass homolog screening but does not pass the molecular network screening? In Figure S1, what are the labels on the network edges, why sometimes there are multiple labels on the same edge?

Response: Thank you for your comment.

(1) The molecular network was constructed based on the GNPS spectra similarity. The internal network was constructed for each homolog series separately. The MS/MS tolerance (0.01Da) and spectra similarity threshold (0.3) have been used. We have revised the statement to clarify, “Briefly, precursor ions differing by CF₂ units were selected as candidates through homolog

screening (Fig. 1b)³³. After spectra purification, the first-layer internal network was constructed for each candidate class based on GNPS spectra similarity (similarity > 0.3, Fig. 1c)³⁴.” (Line 96-99) “The first layer is the homolog internal network, constructed for each potential homolog class separately (Fig. 1c). The node within the internal network exhibited one homolog in the class, whereas the edge of the internal network represented the spectra similarity between nodes. The spectra similarity was calculated using the GNPS spectra similarity algorithm³⁴, which is modified cosine similarity algorithm that considers neutral losses, where the MS/MS tolerance is set to 0.01 Da. Edges with a similarity greater than 0.3 were retained in the internal network. The parameter of 0.3 was assessed based on false positive and false negative rates (Supplementary Fig. 26). Features were removed if they did not have any edge connecting them to other features in the same class. The edge labels represent common fragments between two nodes. Common fragments refer to those with the same mass or differing by CF₂ units. The internal network was used to filter false homologs in potential classes and provide common fragment information of classes to assist in structure annotation.” (Line 368-381)

(2) There are 291 classes after the homolog screening, but only 82 classes after the molecular networking. This is because some candidates match only in terms of parent ions but lack spectral similarity between series. We only performed homolog screening on features with MS2 spectra, considering that identification requires MS2 spectra. We have clarified this, “Homolog screening was conducted for features with MS/MS spectra based on the precursor ion mass difference of 49.99681 Da (CF₂) as previous studies^{33, 51}.” (Line 361-363)

(3) The internal network may lead to false negative. We assessed the false positives and false negatives of the method based on our local mass spectrometry database (which includes 133 perfluoroalkyl substances and 16,734 non-perfluoro substances) and in-house standards. The results indicate that our method reduces 94% false positive features (confirmed by the structures) and increased 11% false negative rates. The standard sample shows that our method has lower false negative rates compared to FluoroMatch and FindPFAS, which employs MS and MS/MS information for prioritization, respectively. The relevant methods and results have been included in the main text. “Our internal network which integrates homolog screening and

molecular network while utilizing both MS1 and MS/MS information, could reduce false positives in homolog screening. We evaluated the denoising capability of the internal network using the local spectra database and an in-house standard sample. Utilizing the local spectra database, we observed that compared to homolog screening, the internal homolog network demonstrated a 94% reduction in false positive features (Supplementary Fig. 1a). The false negative rate increased by 11% after using the internal network (Supplementary Fig. 1b), which may be due to the low spectra similarity caused by different instruments. We also compared our screening results with FluoroMatch 4.5³⁶ and FindPFAS²³ using the standard sample. We found that our approach has fewer false positive and false negative features (Supplementary Fig. 1c) than FluoroMatch. This is reasonable considering that FluoroMatch is based on the homolog screening. Our approach demonstrates comparable false positives and significantly lower false negatives relative to FindPFAS²³, which employs MS/MS spectra information for prioritization. This evaluation suggests that our approach effectively integrates MS1 and MS/MS information, thereby leveraging the strengths of current strategies.” (Line 104-119) “We evaluated the denoising capability of the internal network using a local mass spectra database and an in-house standard sample (100 µg/L). The local spectra database contains 133 PFAS from 45 classes, while the standard samples include 32 PFAS from 6 classes. MS1 tolerance was set at 5 ppm for homolog screening and internal network (Supplementary Fig. 1a). Methanol blanks were used in FluoroMatch³⁶ for filtering and the full-scan peak intensity threshold was set at 10,000, the same as the threshold we used in peak picking. The precursor mass tolerance was set to 0.002 Da, and the MS/MS fragment tolerance was set to 0.01 Da for FluoroMatch. We utilized the "fragment differences" module of FindPFAS for comparative analysis. The fragment difference settings were configured as follows: FindPFAS_1: CF₂, FindPFAS_2: CF₂, C₂F₄, HF, and FindPFAS_3: CF₂, C₂F₄, HF, CF₃, CF₂O. The parameters for these configurations were set to: Number of Differences Desired: 1.0, Fragment Mass Tolerance: 0.01 Da, Intensity Threshold: 5.0, Remove Multiple Mass Tolerance: 0.002, and Occurrence Number Threshold: 20.0. Additionally, we evaluated the analytical method using typical fragments with the “diagnostic fragment” module of FindPFAS. For this evaluation, the fragments were set to the default values, including [CF₃]-, [PO₂F]-, [SO₂F]-, [C₂F₅]-, [C₃F₇]-, [C₄F₉]-, [C₅F₁₁]-, [C₆F₁₃]-, and [C₇F₁₅]-.

False positive features refer to non-PFAS that were incorrectly identified as PFAS, and false

negative features refer to PFAS that were not identified as PFAS. The false positive rate (FPR) was calculated by dividing the number of false positive features by the number of non-PFAS in the database. The false negative rate (FNR) was calculated by dividing the number of false negative features by the number of PFAS in the database. The total number of non-PFAS in the standard samples is difficult to determine. Therefore, the FPR and FNR were not calculated for the standard sample.” (Line 415-438)

Supplementary Figure 1. The comparison of homolog network with other methods. **a** The number of false positive features and false positive rates of homolog screening (HS) and internal network (IN) using the local mass spectrometry database. HS_2 and HS_3 refers to HS with a minimum of 2 homologs per class and 3 homologs per class, respectively. IN_2 and IN_3 refers to IN with a minimum of 2 homologs per class and 3 homologs per class, respectively. **b** The number of false negative features and false negative rates of homolog screening (HS) and internal network (IN) using 100 ug/L PFAS standard sample. **c** The number of false positive features and false negative features of internal network, FluoroMatch, FindPFAS_1 with CF₂, FindPFAS_2 with CF₂, C₂F₄, and HF, and FindPFAS_3 with CF₂, C₂F₄, HF, CF₃, and CF₂O, the diagnostic fragment module of FindPFAS.

(4) We have added two examples of classes that pass homolog screening but did not pass the internal network (Supplementary Figure 3). The labels on the network edges represents the common fragments between the nodes. There might be multiple common fragments between two nodes. Each edge has only one label. Due to the proximity of the edges, it may appear that some edges have multiple labels.

Supplementary Figure 3. Examples of classes that pass the homolog screening but do not pass internal network. **a** Four spectra of the class m/z 441.018868, 491.01419, 591.00897, 691.00128 Da; **b** Two spectra of the class m/z 610.98816, 790.9967 Da.

2. Line 96-118: While I appreciate the visualization from this “external network”, I find it not producing scientific insights. This paragraph is mostly descriptive. I suggest the authors to remove this analysis, or reduce the amount of descriptive texts and focus on the scientific implication from this “external network”. Also, I do not see a clear need for generating a unified spectrum for each PFASs class. Is it possible to just use the MS2 spectra of each PFASs to build a large, unified network?

Response: Thank you for your feedback. We agree with you that this paragraph is mostly descriptive. Therefore, we have removed this text. We have provided the identification process to show the three major functions of the external network:

(1) Similar classes would be clustered together and identified at the same time. “PFAS were identified with the assistance of the external network. The external network clustered structurally similar substances together for simultaneous identification. An example was shown in Figure 3a, fragments of 134.98726 Da, 300.97479 Da and 466.95529 Da were observed in the spectra of community a. The neutral losses of 165.98753 Da and 165.9805 Da between these fragments corresponds to C_3F_6O (165.98533 Da), indicating the presence of multiple ether bonds (Fig. 3a). The sequential neutral loss of C_3F_6O has also been found in the spectra of chloro-perfluoropolyether carboxylates reported in in New Jersey soils³⁸. Therefore, these fragments were annotated as $[C_2F_5O]^-$, $[C_5F_{11}O_2]^-$ and $[C_8F_{17}O_3]^-$. The ultimate neutral loss of 143.98679 Da corresponds to $C_2F_4CO_2$. Finally, Node 23 has been identified as perfluorotriether carboxylic acids (PFTTrECAs). We observed the same fragments of $[C_2F_5O]^-$, $[C_5F_{11}O_2]^-$ and $[C_8F_{17}O_3]^-$ and the extra fragment of $[C_{11}F_{23}O_4]^-$ in the spectra of Node 196, indicating an additional C_3F_6O for Node 196 compared to Node 23. Therefore, Node 196 was identified as perfluoropentanether carboxylic acids (PFTeECAs). Similarly, Node 451 was identified as perfluoropentanether carboxylic acids (PFPeECAs).” (Line 137-152)

(2) Derive the structure of unknown classes from suspect classes. “In addition, the external network groups structurally similar suspect classes with unknown classes, enabling annotations to propagate throughout the network. A series of similar fragments were found in the spectra of community b (Fig. 3b). We identified the fragments of $[C_{11}F_{15}]^-$, $[C_{11}F_{17}]^-$, $[C_{12}F_{17}O]^-$ and $[C_{12}HF_{18}O]^-$ based on the spectra of n:2 FTOHs and n:3 FTCAs in this community. The subsequent fragment of 522.98163 Da in Node 181 were identified as $[C_{12}H_2F_{19}O]^-$. The neutral loss of 112.03623 Da corresponds to H_2F_2 and $C_2H_4CO_2$, indicating the presence of the carboxyl group. The fragment of 89.0231 Da corresponds to $[C_3H_5O_3]^-$, indicating that the oxygen is attached to the β carbon. Finally, node 181 was identified as n:2:3 fluorotelomer ether carboxylic acids (n:2:3 FTECAs).” (Line 153-162)

(3) Aggregate in-source fragments and adducts. “The external network also clusters in-source fragments and neutral losses, preventing duplicate annotations for the same PFAS. The spectra

of Node 36 were the same as those of Node 181 (n:2:3 FTECAs, Fig. 3b). The retention time of all the homologs of these two classes consist within 0.05 min tolerance. After comparing their chromatograms, Node 36 was annotated as the $[M+C_2F_3]$ - adduct of n:2:3 FTECAs. Other nodes of this class except for Node 235 were also identified as in-source fragments and adducts of n:2:3 FTECAs (Supplementary Fig. 4), indicating this class undergoes complex source-internal reactions in the mass spectrometry. Similarly, Node 444 were annotated as $[M+C_2F_3]$ - adduct of Node 451 in the community a. Precursor m/z of Nodes 445 and 450 correspond to $[C_8F_{17}O]$ - and $[C_{11}F_{23}O]$ - and these two nodes were subsequently annotated as in-source fragments of Node 196 and Node 451 through the chromatograms.” (Line 163-174)

Figure 3. Spectra of nodes in community a and b of external network. **a** The part of community a (blue circle) in the external network and MS/MS spectra of nodes in community a. **b** The part of community b (bright red circle) in the external network and MS/MS spectra of nodes in community b.

We also built a large, unified network based on each PFAS according to your suggestions. The network using each PFAS was relatively complex as shown below (similarity>0.3). In contrast, our method treats each PFAS class as a node for identification to avoid redundant connections between homologs.

3. I am very concerned about the low thresholds used for MASST queries especially for PFASs. Please justify your choice of using 2 matching peaks and 0.5 cosine similarity. Previous research has shown that MS2 fragments will not be able to distinguish the numerous PFASs isomers. The level of structural confidence in Figure 3 needs to be clearly communicated. Could you test the FDR of your search using your local databases?

Response: Thanks for the reviewer's comments. We set the threshold to 0.5 takes into account the differences in spectra between different instruments in the database, aiming to improve the detection rate of PFAS. Additionally, perfluoroalkyl fragments have distinct features compared to other compounds, such as negative mass defect values (such as 118.99256 Da, 168.98937 Da), which help reduce false positive rates. We agree with reviewer's concerns and had tested the performance of MASST search under different parameters using PFAS standards samples (35 PFASs), blank sample of menthol (2207 spectra) and local spectra database (830 PFASs and 16734 non-PFASs). We optimized the MASST search parameters in two aspects, 1) mass accuracy settings, including precursor mass error and MS/MS fragment error; 2) spectra match settings, including minimum matched number of fragments and matching score thresholds. Finally, we used the following settings, a single spectrum search was completed using the online

workflow (<https://ccms-ucsd.github.io/GNPSDocumentation/>) on the GNPS website (<http://gnps.ucsd.edu>). The data was filtered by removing all MS/MS fragment ions within +/- 17 Da of the precursor m/z. MS/MS spectra were window filtered by choosing only the top 6 fragment ions in the +/- 50 Da window throughout the spectrum. The precursor ion mass tolerance was set to 0.005 Da and a MS/MS fragment ion tolerance of 0.01 Da. The library spectra were filtered in the same manner as the input data. All matches kept between input spectra and library spectra were required to have a score above 0.5 and at least 2 matched peaks. Detailed evaluations are as follows:

- 1) Mass accuracy settings. We tested the mass accuracy of our mass spectrometer using 35 PFAS standards under concentration 0.5, 1, 2, 5, 10, 20, 50, 100 ug/L. As results shown in Supplementary Figure 27, the precursor mass errors of all PFASs were in ± 0.002 Da. Besides, considering the common mass error of high-resolution mass spectrometer and settings used in other study using MASST (doi.org/10.1038/s41586-023-06906-8, doi.org/10.1038/s41467-022-34537-6), we set the precursor error at 0.005 Da and MS/MS error at 0.01 Da.

Supplementary Figure 27. Precursor mass errors of 35 PFAS standards under

concentration from 0.5-100 ug/L on our instrument.

- 2) Minimum matched number of fragment and matching score threshold. Under precursor mass error 0.005 Da and fragment error 0.01 Da, we evaluated the MASST search algorithm in different parameters. We used 100 ug/L and 1 ug/L sample to test the discovery rate (against 830 local PFASs spectra) and false discovery rate (against 16734 local non-PFASs spectra and 2207 blank sample spectra). We defined discovery rate and false discovery rate as follows in MASST search:

Discovery rate=Number of true PFASs /Number of used PFASs;

False discovery rate= Number of false non-PFASs/ Number of used non-PFASs;

The results are shown in Supplementary Figure 28 a-d. Under similarity threshold 0.5 and minimum fragments 2, 100 ug/L had a 0.625 discovery rate and 1 ug/L had a 0.6 discovery rate, while their false positive rate is 0 under any threshold. It is to be noted that we only use one spectrum for each of the PFAS for both standard sample and local database, thus the spectra come from different instrument and at different collision energy, making the discovery rate lower than 1 at any threshold. Besides, the false positive rates maintain at 0 at all similarity threshold. Due to the number of PFAS used only 35, we expanded the number of searched PFAS using 830 PFAS from local database (Supplementary Figure 28c). The false discovery rate is 0.017 under similarity 0.5 and 2 matched fragments, which is acceptable.

Supplementary Figure 28. MASST search parameters optimization. **a** the false discovery rate and **b** discovery rate of 35 PFAS from 100 ug/L standards, 35 PFAS were search against 830 PFAS, 16734 local non-PFAS spectra and 2207 blank sample spectra. **c** the false discovery rate and **d** discovery rate of 35 PFAS from 1 ug/L standards. **e** the false discovery rate of 830 PFAS, 830 PFAS were search against 16734 local non-PFAS spectra and 2207 blank sample spectra.

We have added these results in maintext and supporting information. “For our final analysis, we utilized the following MASST search settings: a single spectrum search was conducted via the online GNPS workflow (<https://ccms-ucsd.github.io/GNPSDocumentation/>), with data

filtered to exclude all MS/MS fragment ions within ± 17 Da of the precursor m/z. MS/MS spectra were window-filtered to retain only the top 6 fragment ions within a ± 50 Da window across the spectrum. The precursor ion mass tolerance was set to 0.005 Da, and the MS/MS fragment ion tolerance was set to 0.01 Da. Library spectra were filtered similarly to the input data. Matches between input spectra and library spectra required a score above 0.5 and at least 2 matched peaks (1 matched peak was set for n:2 FTSFs considering that only one fragment exist in the spectra of this class). The mass accuracy setting and MS/MS spectrum mating thresholds were validated using PFAS standards (35 PFAS), a menthol blank sample (2,207 spectra), and a local spectra database (830 PFAS and 16,734 non-PFAS) as provided in Supplementary Methods. Searching the 35 PFAS standards and 830 local PFAS spectra against the local non-PFAS database, the false discovery rate is 0 and 0.017 under our search settings, which is acceptable.” (Line 466-480).

MS2 does indeed distinguish structure-similar PFAS isomers, so we have added a description of MASST confidence, “However, MS/MS is often unable to distinguish structurally similar PFAS isomers with similar structures. Therefore, the MASST query results are defined as level 3, which means functional groups can be identified.” (Line 481-483)

Figure 5. **b** Detection of identified PFAS in public MassIVE database, the value refers to the number of the datasets in which each PFAS was detected. PFAS were matched at confidence level 3, indicating that only the presence of functional groups is confirmed, and there may be matches with isomers.

4. I did not find any information on code and data availability. I suggest the authors to make their analysis, raw data, and reference spectra publicly available.

Response: Thank you for your comment. We have uploaded our raw data and code as shown in Code and data availability. “The raw mass spectrometry files of the samples and the local database are available on MASSIVE with the accession number MSV000095719 (doi:10.25345/C5251FX35). The analysis data in the nontarget screening and reference spectra of identified PFAS in mgf files were also provided in Zendo. (<https://zenodo.org/records/13626193>). The code used in this study are available in Zendo

<https://zenodo.org/records/13482320>)” *(Line 492-499)*

Minor comments:

1. Line 20-24, Line 80: In the abstract mentioning types and numbers of samples you used for PFASs discovery and detection. Same applies to the end of Introduction.

Response: Thank you for your suggestions. We have added types and numbers of samples in the abstract and the introduction. “A total of 94 PFAS were detected in twelve waterproof products and two related industrial sludge” *(Line 19-20)* “In this work, we conducted nontarget mass spectrometry on waterproof products and related industrial sludge to identify novel PFAS. The samples analyzed included 2 waterproof chemical samples, 4 rubber car wiper samples, 5 textile samples, 1 waterproof cloth sample, and 2 industrial sludge samples.” *(Line 79-82)*

2. Line 98: It is unclear how the network was divided to 10 communities. Be specific about your parameters.

Response: Thank you for your comment. We have added the parameters, “Finally, community detection was performed on the homolog external network using the Louvain algorithm³⁵, implemented through the community detection module in Gephi. This analysis was conducted with a resolution setting of 1.0 and considered the edge weights, which is the class spectra similarity (S).” *(Line 393-396)*

3. The texts in Figure 1b-d and the sample names in Figure 3a and 3c are not visible.

Response: Thank you for your comment. We have redrawn these figures to make the texts visible.

Figure 1. **a** Workflow of nontarget and suspect analysis. **b** Illustration of homolog screening. The mz_i refers to the precursor m/z of features while mz_{CF_2} refers to the m/z of CF_2 unit. **c** An example to build an internal homolog network to filter false positive homologues. A class containing 7 features (brown circle) were input into the internal network. Four features (blue circle) were retained because they formed edges with other features within this class, while three features (grey circle) were filtered because they did not form any connections within this class. **d** Examples of building an external homologue class network to accelerate unknown PFAS identification. Spectra of each class would be merged into a single spectrum, which served as a node to construct the external network.

Figure 5. **a** Detection of identified PFAS in local database, the shade of the colour represents the peak area. From the inner ring to the outer ring, the columns represent: three air samples, four rice samples, and three wastewater samples, respectively. **b** Detection of identified PFAS in public MASSIVE database, the value refers to the number of the datasets in which each PFAS was detected. PFAS were matched at confidence level 3, indicating that only the presence of functional groups is confirmed, and there may be matches with isomers. **c** The distribution of detected PFAS in the public database. Darker colour indicate a higher number of detected datasets.

4. Figure 2a: what is the inner ring showing?

Response: Thank you for your comment. The inner ring shows the numbers of identified legacy, emerging and novel PFAS class. We have detailed in the figure legend.

Figure 4. Ninety-four identified PFAS from 18 classes detected in waterproof related samples.

a The inner ring represents the number of legacy (light orange), emerging (light blue) and novel

(light green) PFAS class while the outer ring represents the number of legacy (orange), emerging (blue) and novel (green) PFAS. **b** Inferred structures of 18 identified PFAS classes.”

5. Line 208: How did you classify the public datasets into these categories? Through manual inspection?

Response: Thank you for your comment. We classified the public datasets into environment, biota, and human samples manually, using their descriptions and metadata provided by the MASST search tool, which details the origin and associated metadata of the matched MS/MS spectra. This process is now clarified in the revised text: “Finally, the matched datasets were classified into environment, biota and human samples manually based on their species information and descriptions.” *(Line 483-485)*

6. Line 262: How many products were sampled in each category?

Response: Thank you for your comment. We have added the sample numbers in the introduction section, “In this work, we conducted nontarget mass spectrometry on waterproof products and related industrial sludge to identify novel PFAS. The samples analyzed included 2 waterproof chemical samples, 4 rubber car wiper samples, 5 textile samples, 1 waterproof cloth sample, and 2 industrial sludge samples.” *(Line 79-82)*

7. Line 297: Please clarify criteria three here. Did you get rid of fragments from a spectrum if the fragments are present in blank samples? If yes I do not understand the reason to do this filtering.

Response: Thank you for your comment. We initially set this criterion to exclude fragments that appear too frequently in blanks, which could be solvent or background interference. These fragments may influence the effect of molecular network. However, we checked the high frequency fragment lists in blanks and found that some important fragments, such as SO_3^- . Therefore, we removed this criterion considering it might lead to the loss of important fragments and reanalyze our data.

Disposition of Reviewers' comments

REVIEWER COMMENTS

Reviewer #2 (Remarks to the Author):

The author has adequately addressed my comments. I appreciate the author's effort to test the workflow parameters in detail and include results in the Supplementary Information.

Response: Thank you very much for your detailed feedback and constructive comments during the first round of review. We greatly appreciate your thorough review of the various aspects of our work. Your suggestions have not only helped us improve the research but also enhanced the quality of the manuscript.

Reviewer #3 (Remarks to the Author):

The manuscript entitled “Global and Historical Use of Novel Contaminants Revealed by Two-Layer Homolog Network and Database Mining” was thoroughly revised by the authors. It now has a clearer method description, the important parameters from the network analysis are provided and the identification procedures are sufficiently described. Nevertheless, I still see some important issues concerning the transparency and interpretation of the results, especially the robustness of the findings of novel PFAS in the online database, that need to be addressed before publication.

Response: Thank you for your thoughtful feedback and for acknowledging the revisions we made to improve the manuscript. We have implemented several measures to address your concerns and provide greater clarity regarding the methodology and results. (1) We have increased the cosine similarity threshold, clarified the rationale for setting MASST parameters, and added mirror plots and links for each matched PFAS. Additionally, we have discussed the limitations of MASST on isomers and blanks. (2) We provided information on the MS2 coverage and the instrument parameters used to

obtain more MS2 spectra. (3) We discussed details of the method comparison, and moved the methods and discussion of the comparison to the supplementary information. (4) We searched for the structures of novel PFAS on PubChem and SciFinder and included the relevant information. We hope these modifications will enhance the interpretability and transparency of the results.

Major comments:

- The current presentation of the detection of newly identified PFAS in the online database (MassIVE) is questionable! As I understood, the acquired MS2 spectra of newly identified PFAS were matched with MS2 spectra (from HRMS raw data) in the database. The authors used an MS1 tolerance of 5 mDa, 2 (!) matched peaks, and a modified cosine similarity > 0.5. I am highly concerned that the rate of false positives can be high with those parameters. Many PFAS have rather similar MS2 spectra although their structures are different, and they become even more similar when using the “modified cosine similarity” which considers neutral losses and mass shifts. There are clear examples needed that show spectral similarity (mirror plot) between the database spectra and the ones identified in this study! If hits are only on identification level 3, then they might not be the newly identified PFAS but other structures. Furthermore, since HRMS raw data files are present in the database, the MS2 spectra could also come from background signals not originating from the samples. A clearer description of assumptions and limitations is needed!

Response: Thank you for your valuable feedback. We appreciate your concerns regarding the spectral matching process. We would like to clarify the methodology and provide additional information to address your points.

Regarding the parameters, we would like to clarify that the spectral matching in MASST was performed using cosine similarity, not the modified cosine similarity used in the internal network (<https://ccms-ucsd.github.io/GNPSDocumentation/masst/>). We have added this in the Methods section, “Matches between input spectra and library spectra required a cosine similarity score above 0.6 and at least 2 matched peaks” (Line

460-461). The parameters of MS1 tolerance, matched peaks and cosine scores were chosen based on our testing results and related literatures. We have also increased the cosine similarity threshold from 0.5 to 0.6 in order to improve the robustness of the results.

MS1 tolerance: We tested the mass accuracy using 35 PFAS standards across various concentrations (0.5, 1, 2, 5, 10, 20, 50, 100 $\mu\text{g/L}$). Results (shown in Supplementary Fig. 55) indicate that the precursor mass errors were within ± 0.002 Da. Considering the typical mass error of high-resolution mass spectrometers in MassIVE, we set the precursor mass error to 0.005 Da. This threshold setting is lower than those used in MASST-related studies in *Nature* (0.01 or 0.02 m/z for QE Orbitrap and Q-ToF data, doi.org/10.1038/s41586-023-06906-8) and *Nature Communications* (0.01 Da, doi.org/10.1038/s41467-022-34537-6).

Matched peaks: We set the minimum number of matched fragments to 2 because PFAS compounds typically generate fewer fragments compared to metabolites. Statistical analysis of the MS/MS spectra from 94 identified PFAS compounds (Supplementary Fig. 56) revealed that 13% of the PFAS compounds had only two effective fragments. Additionally, we assessed the discovery rate and false discovery rate using samples at concentrations of 100 $\mu\text{g/L}$ and 1 $\mu\text{g/L}$ by searching an in-house PFAS dataset (see Supplementary methods). Our results demonstrated that increasing the minimum number of matched fragments from 2 to 3 led to a 20% reduction in discovery rate at 100 $\mu\text{g/L}$ and a 29% reduction at 1 $\mu\text{g/L}$ (Supplementary Fig. 57 b&d). Moreover, this parameter setting aligns with the approach used in two MassIVE-related studies published in *Nature Communications* (doi.org/10.1038/s41467-022-34537-6).

Cosine scores: We have raised the cosine similarity from 0.5 to 0.6 to improve the robustness of the results, excluding results below this threshold, and we revised Figure 5 and related results accordingly. (Line 272-273) (Fig. 5) We tested the discovery rate at different similarity thresholds using reference standards and found a significant decrease in the discovery rate when the threshold was set above 0.6 (Supplementary Fig. 57 d). To balance the discovery rate and false discovery rate, we ultimately decided to set the threshold at 0.6. We evaluated the false discovery rate by searching 35 PFAS

standards and 830 local PFAS spectra against 16,734 non-PFAS and 2,207 blank features. The false discovery rate was 0 and 0.017 (Supplementary Fig. 57) under a similarity threshold of 0.6 and 2 matched fragments, which is considered acceptable. We have added these in the Supplementary Information. “The parameters of MS tolerance, matched peaks and cosine scores were chosen based on the following reasons. Mass Accuracy Settings: We tested the mass accuracy using PFAS standards. Results (shown in Supplementary Fig. 55) indicate that the precursor mass errors were within ± 0.002 Da. Considering the typical mass error of high-resolution mass spectrometers in MassIVE and settings used in other MassIVE studies (e.g., doi.org/10.1038/s41586-023-06906-8, doi.org/10.1038/s41467-022-34537-6), we set the precursor mass error to 0.005 Da and the MS/MS fragment ion error to 0.01 Da.

Minimum Matched Fragments: PFAS compounds typically generate fewer fragments compared to metabolites. Statistical analysis of the MS/MS spectra from 94 identified PFAS compounds (Supplementary Fig. 56) revealed that 13% of the PFAS compounds had only two effective fragments. Additionally, we assessed the discovery rate and false discovery rate using samples at concentrations of 100 $\mu\text{g/L}$ and 1 $\mu\text{g/L}$ by searching an in-house PFAS dataset (see Supplementary methods). Our results demonstrated that increasing the minimum number of matched fragments from 2 to 3 led to a 20% reduction in discovery rate at 100 $\mu\text{g/L}$ and a 29% reduction at 1 $\mu\text{g/L}$ (Supplementary Fig. 57 b&d). Moreover, this parameter setting aligns with the approach used in two MassIVE-related studies.

Cosine Score Threshold: We tested the discovery rate at different similarity thresholds using reference standards and found a significant decrease in the discovery rate when the threshold was set above 0.6 (Supplementary Fig. 57 d). To balance the discovery rate and false discovery rate, we ultimately decided to set the threshold at 0.6. We evaluated the false discovery rate by searching 35 PFAS standards and 830 local PFAS spectra against 16,734 non-PFAS and 2,207 blank features. The false discovery rate was 0 and 0.017 (Supplementary Fig. 57) under a similarity threshold of 0.6 and 2 matched fragments, which is considered acceptable.” (Supplementary Notes)

Supplementary Figure 55. Precursor mass errors of 35 PFAS standards under concentration from 0.5-100 ug/L on our instrument.

Supplementary Figure 56. Numbers of effective fragments of 94 identified PFAS.

Supplementary Figure 57. MASST search parameters optimization. **a** the false discovery rate and **b** discovery rate of 35 PFAS from 100 ug/L standards, 35 PFAS were search against 830 PFAS, 16734 local non-PFAS spectra and 2207 blank sample spectra. **c** the false discovery rate and **d** discovery rate of 35 PFAS from 1 ug/L standards. **e** the false discovery rate of 830 PFAS, 830 PFAS were search against 16734 local non-PFAS spectra and 2207 blank sample spectra.

We have manually checked the mirror plot and included examples for each matched PFAS in the supplementary information (Supplementary Figs. 26-53), with two examples presented below. Additionally, we have provided links to all matched PFAS in Supplementary Data 3, where the specific matching details can be viewed. “The mirror plots of these matched PFAS are provided in Supplementary Fig. 26-53, and the work links of these matching are available in Supplementary Data 3.” (Line 266-268)

Supplementary Figure 46. Mirror plot of 11:3 FTCA.

Supplementary Figure 48. Mirror plot of 6:2 FTSF.

MASST cannot fully distinguish isomers with the same fragmentation patterns. However, these isomers with similar fragmentation patterns typically also share very similar structures. Furthermore, as you mentioned, these PFAS may indeed originate from backgrounds due to factors such as experimental contamination. This highlights a potential issue with searching public databases, which requires continued collaboration between database maintainers and data contributors to address. However, the detection of PFAS in the background also suggests their potential widespread presence in the local environment. We have also clearly addressed these limitations in the Discussion. “MASST is unable to distinguish between PFAS isomers with same fragmentation patterns, suggesting that the matched results may correspond to structurally similar PFAS. As the matching is conducted on raw data files, the matched PFAS may originate from background blanks rather than the actual samples. However, PFAS in the blanks

also underscores their potential widespread presence in the environment.” (Line 310-315)

- I could not find any information about your MS2 coverage. Was iterative data-dependent acquisition with multiple injections used to get more MS2 spectra? It is stated that 16000 features were detected, but I assume not all of them have MS2 information. As I understood, the network analysis always needs MS2 information. How were features considered that do not have MS2 spectra? Please explain.

Response: Thank you for your comments. Our statement here is ambiguous. Actually, a total of 61,726 features were extracted, of which 16,120 features were assigned MS/MS spectra, resulting in an MS/MS coverage of 26%. We have added this in the manuscript, “A total of 61,726 features were extracted from the mass spectrometry raw data, of which 16,120 features were assigned MS/MS spectra, through peak picking and alignment using MS-DIAL⁴⁰.” (Line 119-121)

We did not use an iterative data-dependent acquisition approach. Instead, we employed the dynamic exclusion with a 6-second exclusion time to obtain more MS/MS spectra. This means that the same ion will be selected for MS/MS analysis only once within 6 seconds, facilitating the analysis of other ions. Additionally, we increased the scan speed for MS/MS acquisition (12Hz) to capture more spectra. We have added these details in the Methods section, “The mass spectrometry data were collected in negative electrospray ionization mode (ESI-), with mass ranges of 80–1000 Da and 50–1000 Da for full scans under mass resolution 70,000 (3Hz) and data dependent MS/MS scans under mass resolution 17,500 (12 Hz), respectively.” (Line 356-359) “The dynamic exclusion was applied to capture more diverse MS/MS spectra, with an exclusion time set to 6 seconds. This allows for each ion to be selected only once for MS/MS analysis within each 6-second interval.” (Line 365-367)

As you mentioned, our network analysis requires MS/MS information. The MS/MS information is also essential for PFAS identification. Therefore, we are currently unable to process features without MS/MS spectra. However, since PFAS in consumer products generally have high concentrations, they are more likely to be

assigned MS/MS spectra, which we deem sufficient for our analysis. We have also addressed this limitation in the Discussion section, where we suggest that future studies could combine iterative DDA or DIA approaches to acquire more MS/MS spectra for better coverage. “Furthermore, PFAS analysis critically depends on MS/MS information, and future investigations could leverage iterative data-dependent acquisition or data-independent acquisition methodologies to improve MS/MS coverage and enhance the robustness of the analysis.” (Line 303-306)

- In general, it is a good idea to make a comparison between existing approaches and your new approach. However, given the very limited amount of data, also here limitations need to be discussed. The comparison between Fluoromatch, FindPFAS, and the author’s approach is only based on a single sample containing 32 PFAS. Those software tools are highly dependent on the chosen parameters and the authors only applied the default parameters without trying to find suitable ones for their data. I would suggest moving some details of the discussion of this comparison into the SI. Furthermore, the authors state that it is difficult to determine the number of non-PFAS in a PFAS standard. Why is that difficult? Do you consider also background compounds in this kind of analysis? If yes, then no reliable FPR rate can be calculated.

Response: Thank you for your feedback. The parameters of these software tools were chosen for specific reasons. In FluoroMatch, there are seven parameters, of which the *Full-scan Intensity threshold*, *m/z Search Tolerance MSI*, *Scan filter in seconds*, and *m/z Search Window MS/MS* settings align with those used in MS-DIAL (our approach). The *MS/MS intensity threshold (file conversion)* and *MS/MS intensity threshold (annotation)* were set to 500, similar to our approach of filtering fragments below 5%. The *Blank Filtering* was set to remove peaks below three times the blank level, consistent with our manual processing. We compared our method with the core module of FindPFAS, specifically the "fragment differences" module. The *Number of Differences Desired* was set to 1, as at least one mass difference is required for the calculation. The *Fragment Mass Tolerance* was set to 0.01 Da, consistent with the other

two methods. The *Intensity Threshold* was set to 5%, the same as our filtering method. The *Remove Multiple Mass Tolerance* was set to 0.002, and the *Occurrence Number Threshold* was set to 20.0, meaning ions occurring more than 20 times would be removed. This had no significant impact on the results, as the same PFAS parent ion in the standard sample could not appear more than 20 times. Additionally, we also tested the performance of FindPFAS under different Fragment Difference settings. We have added these details in the Supplementary Information. “The parameters for FluoroMatch were set to: Full-scan Intensity threshold: 10000, m/z Search Tolerance MS1: 0.002, Scan filter in seconds: 0-3600, m/z Search Window MS/MS: 10 ppm, MS/MS intensity threshold (file conversion): 500, MS/MS intensity threshold (annotation): 500, Blank Filtering: 3. These parameters are similar to or consistent with those used in our approach. We utilized the "fragment differences" module of FindPFAS for comparative analysis. The fragment difference settings were configured as follows: FindPFAS_1: CF₂, FindPFAS_2: CF₂, C₂F₄, HF, and FindPFAS_3: CF₂, C₂F₄, HF, CF₃, CF₂O. The parameters for these configurations were set to: Number of Differences Desired: 1.0, Fragment Mass Tolerance: 0.01 Da, Intensity Threshold: 5.0, Remove Multiple Mass Tolerance: 0.002, and Occurrence Number Threshold: 20.0. These parameters are consistent with our settings or are inconsequential to the results. Additionally, we evaluated the analytical method using typical fragments with the “diagnostic fragment” module of FindPFAS. For this evaluation, the fragments were set to the default values, including [CF₃]-, [PO₂F]-, [SO₂F]-, [C₂F₅]-, [C₃F₇]-, [C₄F₉]-, [C₅F₁₁]-, [C₆F₁₃]-, and [C₇F₁₅]-.” (Supplementary Methods)

However, these evaluations were based on a single sample containing 32 PFAS. Therefore, we have streamlined this section in the main text and moved the detailed method and discussion to the Supplementary Information (Supplementary Notes and Supplementary Methods) according to your suggestions. “We also compared our screening results with FluoroMatch 4.5³⁹ and FindPFAS²⁵ using the standard sample. We found that our approach has fewer false positive and false negative features than FluoroMatch and significantly lower false negatives relative to FindPFAS (Supplementary Fig. 1c).” (Line 112-115) “However, there are certain limitations in this

comparison. Specifically, the parameters for FluoroMatch and FindPFAS were not further optimized to suit the dataset, and the evaluation was based on a limited standard sample.” (Supplementary Notes)

We state that it is difficult to determine the number of non-PFAS in a PFAS standard because each approach employs different peak detection methods, leading to discrepancies in the total number of features detected. As a result, the number of non-PFAS features varies across approaches, making it hard to calculate a reliable false positive rate. However, we can compare the number of false positive features identified by each approach. In fact, the identification efficiency is often determined more by the number of false positive features rather than by the false positive rate itself. We have revised these sentences to clarify, “The total number of non-PFAS features in the standard samples is difficult to quantify due to varying peak detection methods across approaches. Therefore, the FPR and FNR were not calculated for the standard sample.” (Supplementary Notes)

- Please consider also checking PubChem (besides literature) to verify that the structures you have identified are indeed novel. It is difficult these days (with so many publications on PFAS) to check every single PFAS NTS paper and the chance of overlooking novel compounds in literature is high.

Response: Thank you for your suggestions. We would like to clarify that “novel PFAS” used here refer to PFAS that have not been previously detected in samples. However, these PFAS may already be recorded in the PubChem database, as they might be included in patents or other commercial inventories. The presence of these PFAS in PubChem further supports their potential applications. These PFAS, which exist in PubChem but have not been previously reported as detected, are still considered "novel PFAS" in our study because we detected them for the first time. We have revised the sentence to clarify, “Novel PFAS refer to PFAS that have not been previously detected in any matrix” (Line 183-184)

Despite our efforts to thoroughly search recent PFAS nontarget screening articles and reviews, we acknowledge the possibility of omissions. We searched PubChem for each

"novel PFAS" and found that 9 PFAS were recorded in the database, but no related detection reports were available in the literature section. We have added this information in the results, "Among these, C15-PFTECA, C18-PFPeECA, 7:3 FTECA, n:4 FTCAs, and n:2 FTECAs have been recorded in PubChem, indicating their potential commercial use." (Line 209-211)

To further investigate their reporting status, we conducted a SMILES-based search through SciFinder, which revealed one related report of C12-PFTrECA. Accordingly, we reclassified this compound as an "emerging PFAS." We have revised our methods for classifying novel PFAS and updated the related results. "Novel PFAS refer to PFAS that have not been previously detected in any matrix, which was determined based on the list of emerging PFAS compiled from previous literatures and reviews (Supplementary Data 2, updated in August, 2024), as well as references linked to the structure in PubChem and relevant literature retrieved through SciFinder." (Line 183-187) "Seven classes of novel PFAS were discovered, comprising PFTECAs, PFPeECAs, n:3 FTECAs, n:4 FTCAs, dH-PFCAs, H-n:1 FTOHs, n:2 FTECAs." (Line 208-209)

We hope these results could provide a clearer presentation of our findings.

Additional minor comment:

- Abstract: It is hard to understand the approach used for PFAS NTS from the abstract, especially the reduction of false positives by 94%. Please consider explaining this in more detail.

Response: Thank you for your comment. We have revised this part to make it clear, "The first layer constructs molecular networks between homologs, with evaluation showing that it filtered 94% of false homologs. The second layer builds a network between classes to expedite the identification of PFAS." (Line 17-19)

- In my opinion the word "local database" is misleading. If I understood it correctly, this data comes from a sample collection of only 10 samples which is not a database. Please consider changing the wording.

Response: Thank you for your feedback. Our use of “local database” was intended to differentiate it from "public database." However, we understand your concern that this term may not be accurate given the limited number of local samples. Therefore, we have revised it to “local dataset”.

- I noticed several small typos and grammatical mistakes in the manuscript that must be corrected.

Response: Thank you for your comment. We have checked the typos and grammatical errors in the manuscript and made the required corrections. (Line 20, 38, 43, 47, 66, 90, 116, 139, 157, 159, 161, 175, 196, 246, 275, 298, 327, 343, 462, 463)

- I think in the introduction there are still some PFAS prioritization approaches missed that are important. There are other successful approaches besides KMD analysis and diagnostic fragments (e.g., CCS vs. m/z plots or m/C ratios) that can also drastically reduce false positives. Those methods can have even advantages over mass differences since they do not rely on any homologs. In lines 48 – 58, it sounds like other methods cannot identify novel PFAS. Please consider a slight rewriting.

Response: Thank you for your insightful comment. The CCS versus m/z values and m/C ratios are efficient methods for rapid screening of PFAS. The prevalence of fluorine in PFAS results in distinct mass defects (MD), as well as unique CCS versus m/z values and m/C ratios, enabling their separation from hydrocarbon-based pollutants. These methods facilitate fast screening without reliance on homolog series. However, there is still considerable overlap between PFAS and non-PFAS in these values. We have added these methods here, “The collision cross section versus m/z values²² and m/C ratios²³ introduce additional dimensional information, which can partially reduce false positives.” (Line 50-52)

The statement "which hinders the discovery of novel PFAS." was not entirely accurate and could be misleading. We would like to convey that these methods rely on known PFAS fragments, which may limit their ability to identify novel PFAS with different

fragmentation patterns. We have revised this sentence to clarify, “Nevertheless, these features in MS/MS spectra are summarized from known PFAS, which may limit the identification of novel PFAS with different fragment patterns.” (Line 54-55)

- Line 49: “resulting in high false positive rates...”.

Response: Thank you for your comment. We have revised this sentence, “resulting in high false positive rates in complex environmental samples.” (Line49-50)

- Line 65: Shouldn’t it be “MassIVE”?

Response: Thank you for pointing this out. We have revised this sentence and checked this term in the manuscript, “The MassIVE platform reposit millions of...” (Line 67) “The raw mass spectrometry files of the samples and the local database are available on MassIVE” (Line 490-491)

- Line 78: There are many studies on PFAS in waterproof chemicals, especially textiles! Please cite the existing literature properly.

Response: Thank you for your comment. We have reviewed related literatures and found that there are indeed many target studies on PFAS in waterproof-related products, particularly textiles, as well as a few nontarget studies. We have revised these sentences and added relevant literatures, “Extensive targeted studies on PFAS in waterproof-related products, particularly textiles, have been conducted^{32, 33}. Recently, several nontarget studies have also explored the presence of emerging PFAS in textile wastewater and waterproof paper products^{34, 35}, highlighting that waterproof-related products may serve as notable sources of emerging PFAS in the environment.” (Line 75-79)

- Line 127: I would not call that purified, since only signals above a fixed threshold were considered as noise and removed.

Response: Thank you for your comment. To better construct the network, we filtered the fragments with peak intensities below the threshold. The term "purified" could

imply a more extensive process, and we agree with the reviewer that it is not appropriate in this context. Therefore, we have revised the text to use "filtering" instead. The revised sentences are as follows: "After spectra filtering, the first-layer internal network was constructed for each candidate class based on GNPS spectra similarity (similarity > 0.3, Fig. 1c)³⁷." (Line 98-99) "A homolog internal network was constructed for each homolog class after filtering the spectra." (Line 123-124) "We then filtered the MS/MS spectra of these potential PFAS homologs to improve the quality of the network." (Line 375-376)

- Line 142: Isn't this mass error rather high given the specified resolution of the Orbitrap?

Response: Thank you for your comment. The neutral losses mentioned, 165.98753 Da (error +2.20 mDa) and 165.9805 Da (error -4.84 mDa), represent the mass differences between fragments at 134.98726 Da, 300.97479 Da, and 466.95529 Da. The mass errors for these fragments are -0.22 mDa, +1.98 mDa, and -2.86 mDa, respectively. These errors (<3 mDa) are acceptable for MS2 spectra. To achieve a higher scan speed for MS2, we set the MS2 resolution to 17,500 (12 Hz), while the MS1 resolution was set to 70,000 (3 Hz). As a result, the MS2 fragments typically have relatively higher mass errors than precursor ions. We have added the mass errors for these neutral losses and fragments, "The neutral losses of 165.98753 Da and 165.9805 Da between these fragments correspond to C₃F₆O (165.98533 Da) with mass errors of +2.20 mDa and -4.84 mDa, indicating the presence of multiple ether bonds (Fig. 3a)." (Line 135-138) "Therefore, these fragments were annotated as [C₂F₅O]- (error -0.22 mDa), [C₅F₁₁O₂]- (error +1.98 mDa) and [C₈F₁₇O₃]- (error -2.86 mDa)." (Line 140-141)

- **Line 233-239: How was it assessed which classes are dominant? All data is qualitative. Were peak areas used? Please consider how different ionization efficiencies in ESI can be. This needs to be clarified!**

Response: Thank you for your comment. Due to the lack of standards for emerging and novel PFAS, it was challenging to calculate ionization efficiency. Thus, we only compared the peak areas of each class. However, it is true that ionization efficiencies in ESI can vary for different PFAS. Our figure (Supplementary Fig. 25b) might have caused misunderstandings, so we revised the statement and replaced the figure with the raw peak area table (Supplementary Table 2). “We compared the total peak areas of each PFAS class. PFCAs exhibited the highest peak areas in downstream products, followed by PFASs and H-PFCAs (Supplementary Table 2). In contrast, emerging and novel PFAS showed higher peak areas compared to legacy PFAS in the waterproof chemical and sludge samples. The class of n:2:3 FTECAs exhibited the highest peak areas in waterproof chemicals.” (Line 232-236) “BPAFs exhibited the highest peak area proportion in the industrial sludge sample.” (Line 240)

- **Line 290: Waterproof products are a well-known source for PFAS and there are many studies investigating this. Please rewrite the sentence.**

Response: Thank you for your valuable feedback. Indeed, there are many studies investigating PFAS in waterproof products. We have revised the sentence accordingly, “Utilizing this approach, we identified 36 novel PFAS in waterproof-related samples, highlighting previously unreported PFAS in waterproof products.” (Line 290-291)

- **QA/QC section: This section covers only the most basic quality measures. In my**

opinion, if the PFOS background really reaches close to 1 µg/L is very high and concerning. How was the blank correction performed, I could not find a proper explanation. This needs a more detailed discussion!

Response: Thank you for highlighting this. We have evaluated the recoveries and relative standard deviation of methods by spiking standard solution to Florisil. The recovery rates of PFAS ranged from 74.3% to 107.7%, with relative standard deviations within 15%. Additionally, we assessed the instrument's mass accuracy using PFAS standards at different concentrations. The statement "PFOS has been detected in the solvent and procedure blank with a value below 1 µg/L" was made because the lowest calibration standard we injected was 1 µg/L (instrument concentration). However, the actual PFOS peak area observed in the blank sample was approximately 10% of the peak area of the 1 µg/L standard. We performed blank correction by subtracting three times the PFOS peak area detected in the blank sample from the corresponding peak areas in the actual samples. We have revised these sentences to clarify, "Twenty-one PFAS standards, including C4-C14, C16 and C18 perfluoroalkyl carboxylic acids (PFCAs), C6-C10, C12 perfluoroalkane sulfonic acids (PFSAs) were used to evaluate the recoveries, relative standard deviation (RSD), and sensitivity of the instrument. The recoveries and relative standard deviation were evaluated by spiking 20 ng standard to Florisil for each PFAS, and treated as waterproof samples. The recoveries of PFAS ranged from 74.3% to 107.7%, with RSD below 15% (Supplementary Table 4). Instrument sensitivity and mass accuracy were evaluated by injecting standards at 1, 5, 10, 50 and 100 µg/L. All PFAS standards could be detected at 1 µg/L and the precursor mass errors were within ±0.002 Da. A procedure blank and a solvent blank were injected in each batch of analysis. Only PFOS was detected in the solvent and procedure blank, with a peak area approximately one-tenth of that observed for the lowest injected standard (1 µg/L). Blank correction was performed by subtracting three times the PFOS peak area in the blank from the corresponding peak areas in the samples." (Line 475-488)

- Line 496: Several times "Zendo" was written instead of "Zenodo".

Response: Thank you for your careful review. We have corrected all instances of "Zendo" to "Zenodo" in the manuscript, including line 493 and line 495.